Human feeding biomechanics: performance, variation, and functional constraints

http://orcid.org/0000-0002-3882-9354 Ledogar Justin A. 1 2 JLedogar@gmail.com
Dechow Paul C. 3
Wang Qian 3
Gharpure Poorva H. 3
http://orcid.org/0000-0002-1807-4644 Gordon Adam D. 2
Baab Karen L. 4
Smith Amanda L. 2 5
Weber Gerhard W. 6
Grosse Ian R. 7
Ross Callum F. 8
Richmond Brian G. 9 10
Wright Barth W. 11
Byron Craig 12
Wroe Stephen 1
Strait David S. 2 5
1 Zoology Division, School of Environmental and Rural Science, University of New England , Armidale, New South Wales , Australia
2 Department of Anthropology, State University of New York at Albany , Albany, New York , United States
3 Department of Biomedical Sciences, Texas A&M University College of Dentistry , Dallas, Texas , United States
4 Department of Anatomy, Midwestern University , Glendale, Arizona , United States
5 Department of Anthropology, Washington University in St. Louis , St. Louis, Missouri , United States
6 Department of Anthropology, University of Vienna , Vienna , Austria
7 Department of Mechanical & Industrial Engineering, University of Massachusetts , Amherst, Massachusetts , United States
8 Department of Organismal Biology & Anatomy, University of Chicago , Chicago, Illinois , United States
9 Division of Anthropology, American Museum of Natural History , New York, New York , United States
10 Department of Human Evolution, Max Planck Institute for Evolutionary Anthropology , Leipzig , Germany
11 Department of Anatomy, Kansas City University of Medicine and Biosciences , Kansas City, Missouri , United States
12 Department of Biology, Mercer University , Macon, Georgia , United States
Jungers William
Electronic publication date: 2016 Jul 26
Publication date: 2016
Volume: 4
Electronic Location ID: e2242
Received 2016 Mar 17; Accepted 2016 Jun 21
Copyright: © 2016 Ledogar et al.
Copyright year: 2016
Copyright holder: Ledogar et al.
License: This is an open access article distributed under the terms of the Creative Commons Attribution License, which permits unrestricted use, distribution, reproduction and adaptation in any medium and for any purpose provided that it is properly attributed. For attribution, the original author(s), title, publication source (PeerJ) and either DOI or URL of the article must be cited.
License URL: https://creativecommons.org/licenses/by/4.0/

Keywords: Evolution, Loading, Bone strain, Cranium

Funding: National Science Foundation Physical Anthropology HOMINID program NSF BCS 0725219, 0725183, 0725147, 0725141, 0725136, 0725126, 0725122, 0725078 ‘Biomesh’ NSF DBI 0743460 This research was funded by grants from the National Science Foundation Physical Anthropology HOMINID program (NSF BCS 0725219, 0725183, 0725147, 0725141, 0725136, 0725126, 0725122, 0725078), the ‘Biomesh’ grant (NSF DBI 0743460), the EU FP6 Marie Curie Actions MRTN-CT-2005-019564 “EVAN,” a SUNY Albany Dissertation Research Fellowship, and a SUNY Albany GSEU Professional Development Grant. The funders had no role in study design, data collection and analysis, decision to publish, or preparation of the manuscript.

==============================
The evolution of the modern human (Homo sapiens) cranium is characterized by a reduction in the size of the feeding system, including reductions in the size of the facial skeleton, postcanine teeth, and the muscles involved in biting and chewing. The conventional view hypothesizes that gracilization of the human feeding system is related to a shift toward eating foods that were less mechanically challenging to consume and/or foods that were processed using tools before being ingested. This hypothesis predicts that human feeding systems should not be well-configured to produce forceful bites and that the cranium should be structurally weak. An alternate hypothesis, based on the observation that humans have mechanically efficient jaw adductors, states that the modern human face is adapted to generate and withstand high biting forces. We used finite element analysis (FEA) to test two opposing mechanical hypotheses: that compared to our closest living relative, chimpanzees (Pan troglodytes), the modern human craniofacial skeleton is (1) less well configured, or (2) better configured to generate and withstand high magnitude bite forces. We considered intraspecific variation in our examination of human feeding biomechanics by examining a sample of geographically diverse crania that differed notably in shape. We found that our biomechanical models of human crania had broadly similar mechanical behavior despite their shape variation and were, on average, less structurally stiff than the crania of chimpanzees during unilateral biting when loaded with physiologically-scaled muscle loads. Our results also show that modern humans are efficient producers of bite force, consistent with previous analyses. However, highly tensile reaction forces were generated at the working (biting) side jaw joint during unilateral molar bites in which the chewing muscles were recruited with bilateral symmetry. In life, such a configuration would have increased the risk of joint dislocation and constrained the maximum recruitment levels of the masticatory muscles on the balancing (non-biting) side of the head. Our results do not necessarily conflict with the hypothesis that anterior tooth (incisors, canines, premolars) biting could have been selectively important in humans, although the reduced size of the premolars in humans has been shown to increase the risk of tooth crown fracture. We interpret our results to suggest that human craniofacial evolution was probably not driven by selection for high magnitude unilateral biting, and that increased masticatory muscle efficiency in humans is likely to be a secondary byproduct of selection for some function unrelated to forceful biting behaviors. These results are consistent with the hypothesis that a shift to softer foods and/or the innovation of pre-oral food processing techniques relaxed selective pressures maintaining craniofacial features that favor forceful biting and chewing behaviors, leading to the characteristically small and gracile faces of modern humans.

Introduction

Human craniofacial architecture is extreme among living primate species. In particular, modern humans (Homo sapiens) exhibit a tall braincase and a small and short maxilla which distinguishes them from even our closest living relatives, the chimpanzees and bonobos of genus Pan (Fleagle, Gilbert & Baden, 2010). Reductions in the size and prognathism of the face, combined with increases in neurocranial globularity, have also been shown to differentiate modern humans from some extinct members of the genus Homo (Lieberman, McBratney & Krovitz, 2002). Homo exhibits an even more pronounced reduction in the size and robusticity of the facial skeleton, as well as in the size of the postcanine dentition and masticatory muscles (e.g., Robinson, 1954; Rak, 1983; Demes & Creel, 1988), relative to australopiths, an extinct informal group of early hominins from which modern humans are likely to be descended (e.g., Walker, 1991; Wood, 1992; Skelton & McHenry, 1992; Strait, Grine & Moniz, 1997; Strait & Grine, 2004; Kimbel, Rak & Johanson, 2004; Berger et al., 2010). Theories purporting to explain the adaptive significance of masticatory reduction in Homo frequently stress the importance of changes in diet, usually involving a shift to foods that require less extensive intra-oral processing (e.g., Robinson, 1954; Rak, 1983; Brace, Smith & Hunt, 1991; Wrangham et al., 1999; Lieberman et al., 2004; Ungar, Grine & Teaford, 2006; Wood, 2009). However, Wroe et al. (2010) suggest that modern human crania are instead adapted to produce forceful bites, based on their conclusion that the human feeding apparatus is mechanically efficient, requires less muscle force than most other hominoids in order to generate comparable bite reaction forces, and should therefore require a less robust structure. This paper evaluates these two alternatives by comparing feeding biomechanics in modern H. sapiens to that of chimpanzees (Pan troglodytes).

A conventional view of cranial gracilization in the lineage leading to modern Homo states that this process was spurred by the development of stone tool technologies (e.g., Ungar, Grine & Teaford, 2006), as tool use reduces food particle size (Lucas, 2004), allowing a reduced bite force per chew and/or fewer chews per feeding bout (Lucas & Luke, 1984; Agrawal et al., 1997; Zink & Lieberman, 2016). Under this hypothesis, tool use reduces the selective advantage offered by anatomical features that increase muscle force leverage and/or buttress the face against feeding loads. In addition to tool use, increased reliance on meat eating may have played a role in the initial stages of masticatory reduction in early Homo (Lieberman, 2008; Ungar, 2012; Zink & Lieberman, 2016). Further gracilization of the jaws and teeth is hypothesized to have occurred with the advent of cooking, which may have been practiced by H. erectus (Wrangham, 2009; Organ et al., 2011), by reducing masticatory stresses (Lieberman et al., 2004; Lucas, 2004) and increasing digestive efficiency (Wrangham et al., 1999; Carmody & Wrangham, 2009; Carmody, Weintraub & Wrangham, 2011; Groopman, Carmody & Wrangham, 2015). If gracilization in Homo is a consequence of the removal of selection pressure to maintain and resist high magnitude or repetitive bite forces, then human feeding systems should not be optimized to produce high biting forces and the cranium could be structurally weak (i.e., exhibit high stress and strain when exposed to feeding loads).

The hypothesis described above is opposed by an alternative interpretation of human feeding mechanics. A paradox of the human cranium is that the marked facial orthognathism exhibited by recent modern humans increases the mechanical advantage (i.e., leverage) of the muscles responsible for elevating the mandible, allowing humans to generate a given bite force with relatively low muscular effort (Spencer & Demes, 1993; O’Connor, Franciscus & Holton, 2005; Lieberman, 2008; Lieberman, 2011; Wroe et al., 2010; Eng et al., 2013). Many studies interpret bite force efficiency among primate species as being significant in an adaptive sense (Rak, 1983; Strait et al., 2013; Smith et al., 2015a; Ross & Iriarte-Diaz, 2014), with increases in leverage predicted for species that rely on foods that require forceful biting in order to be processed (e.g., hard seeds or nuts). Therefore, high biting leverage among humans seemingly contrasts with the hypothesis that the human craniofacial skeleton has experienced relaxed selection for traits that favor forceful biting and chewing behaviors (e.g., Brace, Smith & Hunt, 1991; Lieberman et al., 2004; Ungar, Grine & Teaford, 2006; Wood, 2009). However, Wroe et al. (2010) present an alternative view based on their analysis of modern human, extant ape, and fossil australopith feeding biomechanics. Using finite element analysis (FEA), Wroe et al. (2010) found that their human finite element model (FEM) was mechanically more efficient at producing bite forces than the other hominoids in their sample. Additionally, they found that the human cranium experienced stresses similar to those in 3 of the 5 other species when models were scaled to the same surface area and bite force, including Pan. Consequently, Wroe et al. (2010) conclude that the human skull need not be as robust in order to generate, or sustain, bite reaction forces comparable to those of other hominoids, and that powerful biting behaviors may have been selectively important in shaping the modern human cranium.

Here, we use FEA to test two opposing mechanical hypotheses: that relative to chimpanzees, the modern human craniofacial skeleton is (1) less well configured, or (2) better configured to generate and withstand high magnitude unilateral bite forces. Our analysis builds on previous research into human craniofacial function (e.g., Lieberman, 2008; Wroe et al., 2010; Szwedowski, Fialkov & Whyne, 2011; Maloul et al., 2012) by examining masticatory biomechanics within the context of the constrained lever model (Greaves, 1978; Spencer & Demes, 1993; Spencer, 1998; Spencer, 1999), which predicts that bite force production in mammals is constrained by the risk of generating distractive (tensile) forces at the working (biting) side temporomandibular joint (TMJ). Under this model, during unilateral biting, reaction forces are produced at the bite point and the working and balancing (non-biting) side TMJs. These three points form a “triangle of support,” and the line of action of the resultant vector of the jaw elevator muscle forces must intersect this triangle in order to produce a “stable” bite in which compressive reaction forces are generated at all three points (Fig. 1A). The resultant vector lies in the midsagittal plane when the muscles are recruited with bilateral symmetry and will pass through the triangle of support during bites on the incisors, canines, and premolars. However, molar biting changes the shape of the triangle such that a midline muscle result may lie outside of the triangle of support. If this occurs, a distractive (tensile) force is generated in the working side TMJ that “pulls” the mandibular condyle from the articular eminence (Fig. 1B). In the case of the mammalian jaw, the soft tissues of the TMJ are well suited to resist compressive joint reaction forces in which the mandibular condyle is being “driven” into the cranium, but they are poorly configured to resist distractive joint forces in which the condyle is being “pulled away” from the cranium (Greaves, 1978). Mammals, including humans (Spencer, 1998), avoid this by reducing the activity of the chewing muscles on the balancing side during bites on the posterior teeth. This draws the muscle resultant vector toward the working side and back within the triangle, but the total muscle force available for biting is reduced, thereby reducing peak bite force magnitudes. Thus, although one might expect that a bite on a distal tooth would produce an elevated bite force due to a short load arm (per a given muscle force), this effect is mitigated by the constraint that the muscle force vector must lie within the triangle of support. A finding that constraints on bite force production were especially strong in humans would be consistent with the hypothesis that the human cranium is poorly configured to generate high unilateral bite forces, and inconsistent with the opposing hypothesis.

Figure 1 The constrained lever model of jaw biomechanics.

During biting, the bite point (b) and the temporomandibular joints on the working side (ws) and balancing side (bs) form a “triangle of support” that changes shape when biting on different teeth. During a premolar bite (A) the resultant vector of the jaw adductor muscles (v) passes through the triangle, producing compression (green circles) at all three points. However, during some molar bites (B) the vector falls outside the triangle when the muscles are being recruited equally on both sides of the head, producing compression at the bite point and bs joint, but distraction (red circle) at the ws joint. The recruitment of the balancing side muscles must be lessened in order to eliminate this distraction, thereby causing the vector to shift its position towards the working side and back into the triangle (yellow arrow).

We further build on previous work by considering intraspecific variation in our analysis of human feeding biomechanics. Our prior work has shown that high degrees of intraspecific variation in cranial shape need not necessarily produce a high degree of intraspecific mechanical variation (Smith et al., 2015b), implying that mechanical patterns are conservative and reflect an underlying common geometry that may be overlain by skeletal traits that can vary without dramatically altering the fundamental mechanical framework of the cranium. A caveat, however, is that Smith et al. (2015b) examined only one species, P. troglodytes. Thus, it has yet to be established if this pattern is generalizable across primates (or other vertebrates). Accordingly, we examined mechanical variation among a sample of geographically diverse human crania found to differ notably in shape.

Materials and methods

Analysis of human cranial shape variation and selection of specimens for FEA

We analyzed FEMs of six crania lying at the extremes of human variation, as well as one “average” specimen found to conform closely to an average shape. To select specimens, we analyzed shape variation within a sample of modern human (H. sapiens) crania using previously collected geometric morphometric (GM) data (Baab, 2007; Baab et al., 2010). We analyzed 85 landmarks collected from a sample of 88 Holocene human crania housed at the American Museum of Natural History (AMNH) (Tables 1 and 2). These included mainly facial landmarks combined with a few that characterize neurocranial shape, corresponding to our focus on facial biomechanics in this study. This sample includes individuals from diverse regions across the globe, and provides a cross-section of populations that differ in cranial robusticity (Baab et al., 2010). Landmark data from these 88 specimens were converted to shape coordinates by Generalized Procrustes analysis (e.g., Bookstein, 1991; Slice, 2005) and analyzed using principal components analysis (PCA). We found that the first three principal components (PCs) described 39% of the shape variation in our sample (Fig. 2). In order to maximize shape-related biomechanical variation in our FEMs, we considered variation from all 88 PCs when selecting specimens to be modeled. We first determined those individuals exhibiting the largest distances from the group centroid (i.e., consensus shape), calculated as Euclidean distance using all 88 PCs (Table 3). From among these individuals, we chose the six specimens that exhibited the largest pairwise distances, excluding insufficiently preserved crania, those missing many teeth, and those unavailable for loan (Table 4). These six “extreme” modern human crania included: one male and one female Khoe-San from South Africa (AMNH VL/2463 and AMNH VL/2470, hereafter referred to as “KSAN1” and “KSAN2”); a male from Greifenberg, Austria (AMNH VL/3878, “BERG”); a female from the Malay Archipelago (AMNH 99/7889, “MALP”); a male from the Tigara culture at Point Hope, Alaska (AMNH 99.1/511, “TIGA”); and a male from Ashanti, West Africa (AMNH VL/1602, “WAFR”). An additional specimen, a Native American male from Grand Gulch, Utah (AMNH 99/7365, “GRGL”), was chosen as an “average” representative of human cranial shape based on its close proximity (i.e., small Euclidean distance) to the group centroid and its availability for loan (see Table 3). Note that this individual was incorrectly transcribed as AMNH 99/7333 by Ledogar (2015).

Table 1 Landmarks used in the geometric morphometric analysis of human craniofacial shape.

Coordinate data on these landmarks were collected by Baab (2007) and Baab et al. (2010). The landmarks chosen for the analysis performed here are a subset of those used by Baab et al. (2010), consisting mainly of facial landmarks. Landmark numbers and descriptions correspond to those in Baab (2007).

Landmark	Number1	
Alare (R, L)	13, 40	
Alveolare	11	
Anterior nasal spine	10	
Anterior pterion (R, L)	24, 51	
Basion	67	
Bregma	5	
Canine-P3 contact (R, L)	116, 125	
Center of mandibular fossa (R, L)	97, 103	
Dacryon (R, L)	16, 43	
Distal M3 (R, L)	121, 130	
Frontomalare orbitale (R, L)	20, 47	
Frontomalare temporale (R, L)	19, 46	
Frontosphenomalare (R, L)	23, 50	
Frontotemporale (R, L)	35, 62	
Glabella	7	
Hormion	68	
Incision	71	
Inferior entoglenoid (R, L)	95, 101	
Inferior zygotemporal suture (R, L)	72, 78	
Infraorbital foramen (R, L)	12, 39	
Inion	1	
Jugale (R, L)	26, 53	
Lambda	3	
Lateral articular fossa (R, L)	96, 102	
Lateral prosthion (R, L)	114, 123	
Lingual canine margin (R, L)	124, 115	
M1-M2 contact (R, L)	119, 128	
M2-M3 contact (R, L)	120, 129	
Malar root origin (R, L)	31, 58	
Mid post-toral sulcus	6	
Midline anterior palatine	70	
Mid-torus inferior (R, L)	21, 48	
Mid-torus superior (R, L)	22, 49	
Nasion	8	
Opisthion	66	
Orbitale (R, L)	18, 45	
P3-P4 contact (R, L)	117, 126	
P4-M1 contact (R, L)	118, 127	
Porion (R, L)	27, 54	
Postglenoid (R, L)	94, 100	
Rhinion	9	
Root of zygomatic process (R, L)	32, 59	
Spheno-palatine suture (R, L)	108, 112	
Staphylion	69	
Stephanion (R, L)	34, 61	
Superior zygotemporal suture (R, L)	25, 52	
Supraorbital notch (R, L)	17, 44	
Temporo-sphenoid suture (R, L)	109, 113	
Zygomaxillare (R, L)	14, 41	
Zygoorbitale (R, L)	15, 42	
Note:

1 Landmark numbers correspond to those in Baab (2007).

Table 2 Geographic distribution of human specimens included in the analysis of craniofacial shape variation.

All specimens are housed at the AMNH.

Region/Population	N	
Aboriginal Australian	9	
Khoe-San, South Africa	3	
China	6	
East Africa	7	
Grand Gulch, Utah	10	
Greifenberg, Carinthia, Austria	6	
Heidenheim, Germany	1	
Kakoletri, Peloponnesus, Greece	1	
Maori, Waitakeri, New Zealand	4	
Mongolia	1	
Point Hope, Alaska	12	
Southeast Asia	12	
Tarnapol, Galicia, Poland	2	
Tasmanian	4	
Tierra del Fuego, Argentina	3	
West Africa	7	

Figure 2 Principal components analysis (PCA) of human craniofacial shape variation.

Panels show (A) PC1 by PC2, (B) PC1 by PC3, and (C) wireframes illustrating craniofacial shape change associated with the first three principal components in right lateral, superior, and frontal views. The left and right columns of wireframes represent the negative and positive ends of each component, respectively, scaled to their respective axes. The 10 unique landmarks with the highest loadings for each component are highlighted using a red ellipse on the midline and right side. A single ellipse was used to circle multiple landmarks if they were located close together. Shape differences toward the positive end of PC 1 include: a vertically shorter face with a more projecting brow ridge, a longer and more projecting palate, a more vertical frontal bone that is narrower at pterion, a vault that is expanded posteriorly, and a lower temporal line at stephanion. Shape differences toward the positive end of PC 2 include: a longer cranium with a wider frontal bone, a vault that is angled more postero-inferiorly, wider orbits and a superiorly shifted nasal aperture, and an antero-posteriorly shorter temporal bone. Shape differences toward the positive end of PC 3 include: higher temporal lines at stephanion, a shorter and more orthognathic subnasal region with a less projecting palate, a more inferiorly positioned TMJ, and a more inferiorly positioned midline cranial base.

Table 3 Human crania sorted by their Euclidean distance from the group centroid.

The first 25 specimens represent the most distant from the group centroid, whereas the bottom row represents an “average” representative of human cranial shape based on its close proximity to the centroid. Values in parentheses represent the distances expressed in units of the mean pairwise distance (0.068), which provides information on how much farther a particular cranium is from the centroid than the mean distance. Specimens are coded here following AMNH catalog numbers.

Specimen	Region/Population	Distance from centroid	
VL/24631	Khoe-San, South Africa	0.1011 (1.49)	
VL/38781	Greifenberg, Austria	0.0939 (1.38)	
99/78891	Malay Archipelago, SE Asia	0.0918 (1.35)	
VL/3818	Greifenberg, Austria	0.0885 (1.31)	
VL/269	Tasmanian	0.0881 (1.30)	
VL/229	Kalmuk, Western Mongolia	0.0876 (1.29)	
VL/408	Mhehe, East Africa	0.0871 (1.28)	
99.1/5111	Point Hope, Alaska	0.0871 (1.28)	
99/8155	Aboriginal Australian	0.0842 (1.24)	
99/6562	Māori, New Zealand	0.0830 (1.22)	
VL/271	Tasmanian	0.0824 (1.22)	
VL/24701	Khoe-San, South Africa	0.0788 (1.16)	
VL/1902	Māori, New Zealand	0.0777 (1.15)	
99.1/490	Point Hope, Alaska	0.0770 (1.14)	
99/8165	Aboriginal Australian	0.0767 (1.13)	
VL/272	Tasmanian	0.0750 (1.11)	
VL3619	Greifenberg, Austria	0.0745 (1.10)	
99/7333	Grand Gulch, Utah	0.0741 (1.09)	
99/8177	Aboriginal Australian	0.0740 (1.09)	
VL/2267	Kakoletri, Greece	0.0733 (1.08)	
VL/1729	Tientsin, China	0.0728 (1.07)	
VL/16021	Ashanti, West Africa	0.0727 (1.07)	
VL/274	Tasmanian	0.0721 (1.06)	
VL/2389	Ashanti, West Africa	0.0721 (1.06)	
99/8171	Aboriginal Australian	0.0720 (1.06)	
99/73651	Grand Gulch, Utah	0.0496 (0.73)	
Note:

1 Specimens selected to be modeled using FEA.

Table 4 Pairwise distances between the six human cranial specimens selected for use in FEA.

Values in parentheses represent the distances expressed in units of the mean pairwise distance (0.068). Specimens are coded here following AMNH catalog numbers.

	VL/2463	VL/3878	99/7889	99.1/511	VL/2470	VL/1602	
VL/2463		0.1634 (1.70)1	0.0938 (0.97)	0.1534 (1.59)1	0.1083 (1.12)	0.1145 (1.19)	
VL/3878			0.1469 (1.52)	0.1304 (1.35)	0.1230 (1.28)	0.1385 (1.44)	
99/7889				0.1526 (1.58)1	0.1178 (1.22)	0.1029 (1.09)	
99.1/511					0.1330 (1.38)	0.1256 (1.30)	
VL/2470						0.1049 (1.09)	
VL/1602							
Note:

1 These represent the greatest pairwise distances in the final sample.

Creation of FEMs from “extreme” and “average” human specimens

Construction of solid models

The seven specimens chosen for analysis were CT-scanned at Penn State’s Center for Quantitative Imaging (pixel size = 0.16 mm) and the 2D digital image stacks were used to create seven solid meshes (Fig. 3) using Mimics v 14.0 (Materialise, Ann Arbor, MI, USA), following the methods outlined by Smith et al. (2015a) and Smith et al. (2015b). Mandibles corresponding to the seven crania (except for BERG and KSAN2, which lacked mandibles; see below) were also scanned so that they could be used to direct muscle force vectors in the loading simulations described below. The crania were solid-meshed at similar densities using tet4 elements (element count: GRGL = 2,118,350; BERG = 1,928,931; KSAN1 = 1,620,112; KSAN2 = 1,392,417; MALP = 1,323,093; TIGA = 2,059,433; WAFR = 1,831,053). Solid meshes were then imported as Nastran (NAS) files into Strand7 (Strand7 Pty Ltd, NSW, Sydney, Australia) FEA software.

Figure 3 Human models analyzed in the current study.

Models include one “average” cranium, GRGL (A) and six “extreme” specimens that differ notably in shape, BERG (B) KSAN1 (C) KSAN2 (D) MALP (E) TIGA (F) and WAFR (G).

We created two sets of human FEMs that differed in their assigned muscle force and bone properties. One set of human FEMs (“ALL-HUM” models) was assigned human properties for bone tissue and masticatory muscle force, whereas chimpanzee properties were applied to the second set (“CHIMPED” models). The ALL-HUM models provide the most realistic assessment of human cranial mechanics, in terms of the predicted strains and bite forces. These models also allow for a more thorough examination of intraspecific variation in humans. In contrast, the CHIMPED models permit direct comparisons between our humans FEMs and our previously analyzed FEMs of chimpanzees and fossil hominins (Smith et al., 2015a; Smith et al., 2015b). These comparisons focus on shape-related differences in mechanical performance that are free of the effects of differences in cranial size and bone material properties. Therefore, the comparisons between the CHIMPED human models and the chimpanzee data from Smith et al. (2015a) and Smith et al. (2015b) most directly address our mechanical hypothesis described above because the hypotheses relate specifically to the mechanical consequences of shape differences.

Material properties of tissues

Human cortical bone material properties assigned to the ALL-HUM models were collected from various locations across the craniofacial skeletons of two fresh-frozen human cadavers (female, aged 22; male, aged 42) by measuring their resistance to ultrasonic wave propagation (Ashman et al., 1984; Peterson & Dechow, 2002; Schwartz-Dabney & Dechow, 2002; Wang & Dechow, 2006; Wang, Strait & Dechow, 2006; see Supplemental Information). Previous studies show that freezing has only a very minimal effect on ultrasonic measurements and elasticity of cortical bone (Zioupos, Smith & Yuehuei, 2000). For each location sampled, the elastic (Young’s) modulus in the axis of maximum stiffness (E3) was averaged between the human donors and used to distribute spatially heterogeneous isotropic material properties throughout the seven human FEMs using a method (Davis et al., 2011) analogous to the diffusion of heat through a highly conductive material. To achieve this, values at each of the sampled locations, which ranged from 17.92–25.52 GPa (mean = 20.61 GPa, SD = 1.92), were converted to temperatures and distributed throughout the cortical volume of the FEM. The elastic modulus of cortical bone was then set to vary with temperature during the subsequent loading analysis, with any thermally-induced strains removed from the analysis. For Poisson’s ratios, models were each assigned the average of the sampled locations (v23 = 0.293). The same procedure was used to diffuse chimpanzee material properties to the CHIMPED model variants using data collected from a cadaveric female chimpanzee at 14 craniofacial regions (Smith et al., 2015a; Smith et al., 2015b). In both the ALL-HUM and CHIMPED sets of model variants, homogeneous isotropic properties were used to model both trabecular bone (E3 = 637 MPa; v23 = 0.28) and enamel (E3 = 80,000 MPa; v23 = 0.28), following Smith et al. (2015a) and Smith et al. (2015b).

Muscle forces and constraints

Jaw adductor muscle forces were applied to both sets of FEMs for the anterior temporalis, superficial masseter, deep masseter, and medial pterygoid under the assumption that the chewing muscles were acting at peak activity levels on both sides of the cranium. These loads allow an estimate of the maximum bite force produced by each individual. In the ALL-HUM variants, muscle forces were applied based on muscle physiological cross-sectional area (PCSA) data reported by van Eijden, Korfage & Brugman (1997), with forces corrected to account for pennation and differences in gape during fixation using formulae from Taylor & Vinyard (2013). Corrected PCSAs were then used to calculate forces in Newtons (N) such that each cm2 of muscle was equivalent to 30 N (Murphy, 1998). These unscaled forces were applied to the “average” specimen (GRGL), while the six “extreme” variants were applied forces that were either scaled up or down based on differences in model size (Table 5), with size represented by model volume (i.e., the summed volume of all tet4 elements in mm3) to the two-thirds power. This muscle force scaling procedure removes the effects of differences in model size on stress, strain, and strain energy density (SED) from the mechanical results (Dumont, Grosse & Slater, 2009; Strait et al., 2010). The CHIMPED model variants were also assigned forces that were scaled dependent on their size using PCSA data from an adult female chimpanzee (Strait et al., 2009; Smith et al., 2015a; Smith et al., 2015b). However, rather than scaling the FEMs around the “average” specimen (GRGL), we scaled the forces applied to the CHIMPED models (see Table 5) from the baseline chimpanzee model used for scaling purposes (PC1+) in the analysis by Smith et al. (2015b), permitting size-free comparisons between humans and chimps. For both sets of muscle loadings, plate elements modeled as 3D membrane were “zipped” at their nodes to the surface faces of tet4 elements representing each muscle’s origin. The scaled muscle forces for each set of analyses were applied using Boneload (Grosse et al., 2007) to the normal surfaces of the plate elements as tractions directed toward their respective insertions on the mandible, with the mandible slightly depressed and the condyles translated onto the articular eminences (Dumont et al., 2010). Mandibles were only used here to direct these vectors. In the case of the BERG specimen, which was lacking its mandible, a scaled version of the GRGL mandible was used to define the orientation of muscle force vectors. Similarly, a scaled version of the KSAN1 mandible was used to replace the missing mandible in KSAN2.

Table 5 Muscle force scaling for the ALL-HUM and CHIMPED models of modern human crania.

Muscle forces in Newtons (N) were scaled by model size, where size is represented by model volume in mm3. Models are shown here ordered from smallest to largest in size.

Variant	Model	Volume (mm3)	Volume2/3	Muscle force (N)	
AT	SM	DM	MP	
ALL-HUM	KSAN2	331,466	4,789.53	128.41	105.15	53.29	108.64	
MALP	364,129	5,099.22	136.72	111.95	56.73	115.67	
KSAN1	433,331	5,726.38	153.53	125.72	63.71	129.89	
WAFR	475,555	6,092.57	163.35	133.75	67.79	138.20	
BERG	489,588	6,211.84	166.55	136.37	69.11	140.90	
GRGL	557,223	6,771.52	181.55	148.66	75.34	153.60	
TIGA	655,320	7,544.59	202.28	165.63	83.94	171.14	
CHIMPED	KSAN2	331,466	4,789.53	556.13	572.02	85.07	189.02	
MALP	364,129	5,099.22	592.09	609.00	90.57	201.24	
KSAN1	433,331	5,726.38	664.91	683.90	101.71	225.99	
WAFR	475,555	6,092.57	707.43	727.64	108.22	240.44	
BERG	489,588	6,211.84	721.28	741.88	110.34	245.15	
GRGL	557,223	6,771.52	786.26	808.73	120.28	267.24	
TIGA	655,320	7,544.59	876.02	901.05	134.01	297.74	
Note:

AT, anterior temporalis; SM, superficial masseter; DM, deep masseter; MP, medial pterygoid.

For both sets of biting simulations, each of the seven FEMs was oriented such that one of three axes (i.e., X, Y, or Z) was parallel to the occlusal plane. Each model was constrained at a single node against translation in all axes at the working-side TMJ, while the balancing-side TMJ was constrained only in the superoinferior and anteroposterior directions (Strait et al., 2009; Smith et al., 2015a; Smith et al., 2015b), thus creating an axis of rotation around the TMJs. Models were subjected to simulations of left premolar (P3) and left molar (M2) biting by constraining a node in the center of occlusal surface in each tooth, respectively, in the superoinferior direction. These constraints generated strains in the craniofacial skeleton, as well as reaction forces at the TMJs and bite point, upon the application of muscle forces.

Analysis of model output parameters

Following Smith et al. (2015a) and Smith et al. (2015b), we displayed global strain patterns using strain maps. These maps are analogous to histograms in that they illustrate strain magnitudes at thousands of nodes simultaneously, but have the added advantage of preserving spatial information. In addition, we collected strain data generated by each FEM from surface elements at 14 locations across the craniofacial skeleton (Fig. 4). These locations correspond to those included in previous in vitro and in silico (e.g., FEA) studies on primate feeding biomechanics (e.g., Hylander, Johnson & Picq, 1991; Hylander & Johnson, 1997; Ross et al., 2011; Smith et al., 2015a; Smith et al., 2015b). At each location, we examined several strain metrics from each of the seven FEMs in order to understand patterns of deformation. These included maximum principal strain (tension), minimum principal strain (compression), maximum shear strain (maximum principal strain–minimum principal strain), von Mises strain (distortional strain or non-isometric strain), and strain energy density (SED, the strain energy stored at a given point). Additionally, strain mode, the absolute value of maximum principal strain divided by minimum principal strain, was recorded for each location. This measure indicates whether tension or compression is dominant at a given location.

Figure 4 Key to locations where strains were sampled in FEMs.

Strain data were collected from ALL-HUM and CHIMPED variants of human FEMs from 14 craniofacial sites, following Smith et al. (2015a) and Smith et al. (2015b).

Data on the reaction forces generated at constrained nodes (i.e., the bite point and two TMJs) were recorded in Newtons (N). Reaction forces at the P3 and M2 were recorded relative to the occlusal plane, while reaction forces at the left and right TMJs were recorded and compared relative to a user-defined “triangle of support” Cartesian coordinate system, with one of three axes perpendicular to a reference plane defined by the “triangle of support” formed by the constrained nodes at the bite point and two articular eminences (Smith et al., 2015a; Smith et al., 2015b). The efficiency of bite force production at a given bite point in each model was also compared using the mechanical advantage (MA), a measure of masticatory muscle efficiency or leverage, calculated as the ratio of bite force output to muscle force input.

In the evaluation of our mechanical hypothesis, we first inspected data collected from the ALL-HUM models for large levels of intraspecific variation that could potentially invalidate the functional significance of our results. Strain magnitudes and SED at each of the 14 sampled locations were examined for large differences between individuals, in addition to a comparison of coefficients of variation (CVs) at specific locations. Differences in the spatial patterning of strain magnitudes between the ALL-HUM models were also compared using strain maps, in addition to variation in biting efficiency (i.e., MA). Lastly, we also calculated CVs for von Mises strain and MA in the CHIMPED model variants for direct comparison with the chimpanzee CVs reported by Smith et al. (2015b) using the Fligner-Killeen test for equal CVs.

To analyze relative mechanical performance in our human FEMs, we focused on comparisons between the CHIMPED humans and our previously analyzed FEMs of chimpanzee crania (Smith et al., 2015b). Specifically, we compared the magnitudes of von Mises strain, considered to be a key metric in assessing regional bone strength (Keyak & Rossi, 2000), at the 14 sampled locations, as well as differences in biting efficiency, between humans and chimps. We tested for significant differences between species using the Mann-Whitney U test.

In vitro validation of specimen-specific human cranial FEM

Data on in vitro bone strain collected during simulated P3 biting in a cadaveric human head were used to validate our results. As noted above, two human heads were used to gather data on the properties of craniofacial cortical bone. Before the removal of bone samples, the male specimen was CT-scanned, and strain data from 14 craniofacial locations were collected during a series of in vitro loading analyses (see Supplemental Information). Digital images of the specimen were then used to construct an eighth FEM, the in vitro loadings were replicated using FEA, and strain data were collected from the FEM at locations corresponding to the 14 gage sites. The in vitro and in silico strain data were then compared in order to establish the degree to which assumptions regarding geometry and material properties introduce error into an FEM, where error is represented by the differences between the in vitro (observed) and in silico (expected) results, divided by the expected results. These data were also analyzed using ordinary least squares (OLS) regression. Lastly, the orientations for both maximum and minimum principal strain in FEM were visually compared to those recorded during the in vitro loadings.

Results

In vitro validation of specimen-specific human cranial FEM

Strain magnitudes recorded during in vitro P3 loadings of the human cadaveric specimen and the results of the specimen-specific FEA are listed in Table 6. Comparisons of these data reveal that the specimen-specific FEM generated strains very similar in magnitude to those generated during the in vitro loadings. Results of the regression analysis on log-transformed strain data confirm a close correspondence between in vitro and in silico results, with significant regressions of 0.845x + 0.194 (r2 = 0.909, p < 0.001) and 0.849x + 0.186 (r2 = 0.953, p < 0.001) for maximum principal strain and minimum principal strain, respectively. However, assumptions regarding geometry and material properties did introduce error into the FEM (see Table 6). Visual inspection of principal strain orientations in the specimen-specific FEA reveals that orientations for both maximum principal strain and minimum principal strain at the 14 sampled locations were also very similar to those recorded from the 14 gage locations during the in vitro analysis (Figs. S3–S7).

Table 6 Results of in vitro validation analysis.

Average values and standard deviations for maximum (MaxPrin) and minimum (MinPrin) principal strain magnitudes recorded during three in vitro loading trials on the left P3 biting, the results of a specimen-specific in silico (FEA) loading analysis, and an estimate of the error in the FEA, where “error” is represented by the difference between in vitro (observed) and in silico (expected) results, divided by the expected results. See Figs. S3–S7 for site locations. Units are in microstrain (μɛ).

Site	Exp.	MaxPrin	MinPrin	
1	In vitro	15.00 (4.36)	−10.33 (2.08)	
In silico	14	−15	
Error	6.67%	45.16%	
2	In vitro	13.00 (1.00)	−11.67 (0.58)	
In silico	10	−10	
Error	23.08%	14.29%	
3	In vitro	3.33 (0.58)	−5.00 (1.00)	
In silico	6	−7	
Error	80.00%	40.00%	
4	In vitro	30.67 (1.15)	−36.00 (0.00)	
In silico	29	−34	
Error	5.43%	5.56%	
5	In vitro	15.00 (2.00)	−14.67 (1.53)	
In silico	19	−12	
Error	26.67%	18.18%	
6	In vitro	11.67 (0.58)	−7.33 (0.58)	
In silico	11	−10	
Error	5.71%	36.36%	
7	In vitro	42.33 (1.53)	−23.33 (2.25)	
In silico	42	−17	
Error	0.79%	27.14%	
8	In vitro	42.33 (2.08)	−109.67 (3.06)	
In silico	37	−105	
Error	12.60%	4.26%	
9	In vitro	7.67 (0.58)	−2.67 (2.08)	
In silico	8	−4	
Error	4.35%	50.00%	
10	In vitro	45.33 (2.08)	−22.33 (1.15)	
In silico	23	−20	
Error	49.26%	10.45%	
11	In vitro	23.67 (0.58)	−10.67 (3.06)	
In silico	22	−13	
Error	7.04%	21.88%	
12	In vitro	108.00 (2.65)	−281.67 (8.33)	
In silico	115	−238	
Error	6.48%	15.50%	
13	In vitro	38.67 (1.15)	−22.00 (1.00)	
In silico	39	−17	
Error	0.86%	22.73%	
14	In vitro	27.67 (2.08)	−42.33 (3.01)	
In silico	38	−25	
Error	37.35%	40.94%	

Shape-related variation in human feeding biomechanics

Variation in strain magnitude and spatial patterning

Box-plots of strain and SED distributions recorded from the ALL-HUM models at the 14 sampled locations during premolar (P3) and molar (M2) biting are shown in Fig. 5 (see also Tables S1 and S2). Despite notable differences in craniofacial morphology between the models, comparisons of strain magnitudes reveal strong similarities. For P3 biting, the highest strain magnitudes were experienced at the working nasal margin (Location 12), although on average higher tensile strain magnitudes were generated at the working and balancing postorbital bars (Locations 4 and 5). During M2 biting, the working zygomatic root (Location 8) was subjected to the highest strain magnitudes, except that tension was greatest at the balancing postorbital bar. During both bites, low strain magnitudes were generated along the supraorbital torus (Locations 1–3), the balancing zygomatic root (Location 9), balancing infraorbital (Location 11), and the zygomatic bodies (Locations 13 and 14). All FEMs of human crania were found to exhibit this general pattern.

Figure 5 Strain and SED generated by the ALL-HUM models.

Box-and-whisker plots show the minimum, first quartile, median, third quartile, and maximum for strain and SED magnitudes (y-axis) generated by the ALL-HUM models at the 14 sampled locations (x-axis) during premolar (P3) and molar (M2) biting. Site numbers follow Fig. 4.

Some regions of the face did exhibit large differences among individuals. In particular, the FEMs were found to differ in von Mises strain magnitude by as much as 210% at the nasal margin, which also has the highest CVs for all forms of strain during both P3 and M2 biting (Table 7), with the exception of minimum principal strain at the working dorsal orbital (Location 2) and balancing infraorbital (Location 11) during P3 biting, SED at the working dorsal orbital (Location 2) during P3 biting, and the balancing zygomatic body (Location 14) for both bites.

Table 7 Variation in strain and SED in the ALL-HUM models.

Coefficients of variation for maximum principal strain (MaxPrin), minimum principal strain (MinPrin), shear strain (Shear), von Mises strain, and SED at the 14 locations examined during premolar (P3) and molar (M2) biting in the ALL-HUM models of modern human crania. Site numbers follow Fig. 4.

Site	Bite	MaxPrin	MinPrin	Shear	von Mises	SED	
1	P3	56.01	34.39	28.49	27.88	59.08	
M2	43.20	28.62	20.78	22.82	50.07	
2	P3	28.35	41.61	30.51	29.27	78.82	
M2	27.61	44.20	29.50	29.04	60.38	
3	P3	23.83	26.53	22.94	22.97	52.39	
M2	25.16	24.29	24.66	24.16	49.48	
4	P3	15.30	21.39	14.75	14.28	27.78	
M2	34.43	22.83	22.73	21.46	36.89	
5	P3	14.32	13.06	12.77	13.24	26.98	
M2	12.50	14.22	11.70	12.06	24.53	
6	P3	21.74	12.21	11.77	11.89	23.52	
M2	17.43	13.56	11.13	12.05	25.11	
7	P3	12.53	8.26	8.09	7.93	15.97	
M2	11.27	6.05	5.78	5.32	11.98	
8	P3	19.73	2.58	13.87	12.50	25.96	
M2	20.48	12.04	12.62	11.88	23.36	
9	P3	20.78	21.84	18.18	19.30	39.77	
M2	12.59	9.28	8.23	8.66	19.36	
10	P3	11.70	33.05	12.32	11.72	21.21	
M2	35.51	22.16	25.60	25.86	50.44	
11	P3	24.44	37.84	24.15	21.83	36.54	
M2	25.53	43.20	28.88	26.73	52.39	
12	P3	51.04	35.54	39.39	37.44	64.43	
M2	52.66	34.33	41.78	40.46	76.44	
13	P3	28.41	34.42	26.48	25.60	51.87	
M2	14.11	20.80	14.37	13.50	28.05	
14	P3	35.54	22.56	31.16	31.33	68.31	
M2	39.93	26.73	35.19	35.33	80.97	

Strain mode was nearly always compressive or tensile at a given location across the seven ALL-HUM models (Fig. 6), with a few exceptions. During premolar biting, only 3 locations varied with respect to strain mode (Locations 1, 10, 11), with only one FEM differing from the other models in each case. These three locations also differed in strain mode during molar biting, with Locations 1 and 10 exhibiting slightly higher levels of variation, in addition to variation in strain mode at Location 4.

Figure 6 Strain mode in the ALL-HUM models.

Distribution of strain mode (log of ratio of maximum to minimum principal strain, y-axis) plotted by location (x-axis) in the ALL-HUM models. Plots show (A) premolar (P3) and (B) molar (M2) biting. Logging the data listed in Tables S2 and S3 centers strain mode data around zero. Values above zero indicate mainly tension, while values below zero indicate mainly compression. Site numbers follow Fig. 4.

By comparison with CHIMPED FEMs, humans were found to exhibit lower levels of shape-related variation in von Mises strain magnitude and lower CVs than chimpanzees at the 14 sampled locations (Table 8). However, results of the Fligner-Killeen tests reveal that only 3 of the 14 “gage sites” exhibit significant differences in CV values. Specifically, humans were found to exhibit a significantly lower CV at the zygomatic arches during both P3 and M2 biting, and at the working infraorbital during P3 biting.

Table 8 Variation in von Mises strain magnitudes: Human vs. Chimpanzee.

Comparisons of the coefficients of variation (CVs) for von Mises strain recorded in the CHIMPED human models and the chimpanzee results from Smith et al. (2015b) at each of the 14 craniofacial sites examined. Results of Fligner-Killeen tests for equal CVs between the species are also presented (α = 0.05). Comparisons that yielded significant results are marked by asterisks.

Site		P3	M2	
1	CV–Human	29.04	22.68	
CV–Chimp	25.91	23.63	
p (same CV)	0.065	0.141	
2	CV–Humans	24.34	23.05	
CV–Chimps	46.61	47.07	
p (same CV)	0.122	0.050	
3	CV–Humans	19.71	17.75	
CV–Chimps	19.81	20.10	
p (same CV)	0.386	0.369	
4	CV–Humans	13.51	21.12	
CV–Chimps	29.98	33.20	
p (same CV)	0.176	0.359	
5	CV–Humans	12.89	11.50	
CV–Chimps	27.56	29.40	
p (same CV)	0.156	0.060	
6	CV–Humans	18.15	16.51	
CV–Chimps	64.99	66.99	
p (same CV)	0.022*	0.022*	
7	CV–Humans	11.96	12.07	
CV–Chimps	55.83	56.63	
p (same CV)	0.022*	0.022*	
8	CV–Humans	10.14	12.27	
CV–Chimps	16.54	25.58	
p (same CV)	0.143	0.130	
9	CV–Humans	14.12	8.03	
CV–Chimps	25.7	23.58	
p (same CV)	0.069	0.052	
10	CV–Humans	8.8	15.46	
CV–Chimps	17.36	15.30	
p (same CV)	0.039*	0.290	
11	CV–Humans	10.6	14.34	
CV–Chimps	27.76	28.11	
p (same CV)	0.056	0.100	
12	CV–Humans	38.05	38.76	
CV–Chimps	28.23	43.35	
p (same CV)	0.147	0.396	
13	CV–Humans	24.54	10.39	
CV–Chimps	17.95	17.52	
p (same CV)	0.157	0.207	
14	CV–Humans	22.78	23.11	
CV–Chimps	51.99	55.84	
p (same CV)	0.222	0.166	

Variation in the spatial patterning of strain concentrations

Despite some large differences in strain magnitude, the spatial patterning of strain distributions was similar across the ALL-HUM models. The color maps during P3 biting (Fig. 7) reveal two predominant deformation regimes that are common across the seven FEMs: (1) superior displacement of the anterior maxilla in proximity to the loaded P3, which creates highly tensile and compressive (hence highly shearing) strains surrounding the root of the nasal margin, compression along the nasal margin, and compression at the working zygomatic root; and (2) frontal bending of the zygomae under the inferiorly directed pulling action of the masticatory muscles, which generates tension at the zygomatic body and near the zygomaticomaxillary junction, particularly at the working-side, and deforms the orbit such that it is tensed along an inferolaterally-oriented axis and compressed along a superolaterally-oriented axis.

Figure 7 Strain distributions in the ALL-HUM models: P3 biting.

Color maps of strain distributions in the ALL-HUM variants of “extreme” and “average” modern human cranial FEMs during premolar (P3) biting. Scales are set to range from −150–150 μɛ for both maximum principal strain (MaxPrin) and minimum principal strain (MinPrin), from 0–300 μɛ for both maximum shear strain (Shear) and von Mises strain (von Mises), and from 0–0.5 J/mm3 for SED. White regions exceed scale. Models are shown at the same height.

The color maps of strain patterning during M2 biting were also generally similar across the ALL-HUM models (Fig. 8). As expected, all models exhibited lower strain magnitudes in the lower maxillary region during molar biting compared to premolar biting, but higher concentrations of compressive strain at the working zygomatic root. Molar biting was also associated with the same type of frontal bending, zygomatic torsion, and orbital deformation that was observed for premolar biting, with relatively large concentrations of strain at the postorbital bars, orbital margins, and medial infraorbital.

Figure 8 Strain distributions in the ALL-HUM models: M2 biting.

Color maps of strain distributions in the ALL-HUM variants of “extreme” and “average” modern human cranial FEMs during molar (M2) biting. Scales are set to range from −150–150 μɛ for both maximum principal strain (MaxPrin) and minimum principal strain (MinPrin), from 0–300 μɛ for both maximum shear strain (Shear) and von Mises strain (von Mises), and from 0–0.5 J/mm3 for SED. White regions exceed scale. Models are shown at the same height.

In their study of chimpanzee biomechanical variation, Smith et al. (2015b) compared color maps of principal strain magnitudes in their six models with the scales normalized to an average of 10 landmarks (Locations 1–5, 8–12). They suggest that, by illuminating similarities and differences between individuals in the concentrations of relatively high and low strain concentrations through this normalization step, such “relative strain” maps strain may be particularly informative in comparative analyses of craniofacial function. When viewed in this manner (Fig. 9), the CHIMPED human models more clearly reveal a shared pattern of facial deformation that differs from that of chimpanzees under identical loading conditions, which was predominantly characterized by torsion of the zygoma and resulting orbital deformation under the inferiorly-directed masseteric muscle force.

Figure 9 Relative strain distributions.

Color maps of “relative” maximum (MaxPrin) and minimum (MinPrin) principal strains in the CHIMPED model variants during premolar (P3) and molar (M2) biting. The scales range from −x¯ to x¯, where x¯ differs in each image as follows: P3, MaxPrin/MinPrin: GRGL, 612/644; BERG, 500/534; KSAN1, 508/603; KSAN2, 593/724; MALP, 520/610; TIGA, 455/498; WAFR, 672/742; M2, MaxPrin/MinPrin: GRGL, 505/546; BERG, 468/525; KSAN1, 441/473; KSAN2, 505/546; MALP, 433/458; TIGA, 419/420; WAFR, 530/553. White regions exceed scale.

Variation in bite force production and efficiency

The ALL-HUM models exhibit moderate differences in bite force production and efficiency (mechanical advantage, MA) at P3 and M2 bite points (Table 9). With respect to bite force production, humans generated premolar bite forces that ranged from 333–507 N when loaded with scaled masticatory muscle forces. The MA range for premolar biting was 0.34–0.43 with all but one individual (WAFR) occupying a narrower range of 0.39–0.43. Molar bite forces ranged from 496–756 N. In terms of leverage, most FEMs exhibited molar MAs of 0.57–0.64, but with the WAFR model again being considerably less efficient (0.53).

Table 9 Bite force production, biting efficiency, and joint reaction forces in the ALL-HUM model variants of human crania.

Bite force (BF), mechanical advantage (MA), working-side TMJ reaction force (RF-WS), and balancing-side TMJ reaction force (RF-BS) for premolar and molar biting. Five of seven ALL-HUM models generated distractive (tensile) reaction forces during molar loading. Therefore, balancing side muscle forces were iteratively reduced by 5% and re-run until distractive forces were eliminated. Bite forces and TMJ reaction forces are in Newtons (N).

Model	Muscle force	Premolar bite	Molar bite	
BF	MA	RF-WS	RF-BS	BF	MA	RF-WS	RF-BS	
GRGL	1,118	441	0.39	167.42	349.25	658	0.59	−11.74	329.79	
GRGL1	1,090					642	0.59	−1.37	311.18	
GRGL2	1,062					625	0.59	8.98	292.58	
BERG	1,026	439	0.43	147.72	281.55	663	0.65	−6.98	249.09	
BERG1	1,000					647	0.65	1.29	234.72	
KSAN1	946	378	0.40	121.76	295.69	538	0.57	−17.49	280.57	
KSAN12	898					511	0.57	0.07	249.74	
KSAN2	791	333	0.42	106.83	240.30	496	0.63	−18.86	222.80	
KSAN22	751					471	0.63	−4.26	197.88	
KSAN23	732					459	0.63	3.04	185.41	
MALP	842	344	0.41	131.09	277.66	537	0.64	−19.85	274.49	
MALP2	800					510	0.64	−0.99	242.97	
TIGA	1,246	507	0.41	187.96	373.24	756	0.61	13.68	336.84	
WAFR	1,006	341	0.34	149.36	298.77	529	0.53	12.64	273.79	
Notes:

1 Model re-run using muscle forces reduced by 5% on the balancing side.

2 Model re-run using muscle forces reduced by 10% on the balancing side.

3 Model re-run using muscle forces reduced by 15% on the balancing side.

When compared to the chimpanzee data in Smith et al. (2015a), the CHIMPED human models analyzed here were found to exhibit somewhat lower ranges of variation in biting MA. However, results of the Fligner-Killeen tests reveal no significant differences in CV values between the species at either the P3 (chimp = 8.67, human = 5.65; p = 0.18) or M2 (chimp = 8.11, human = 6.67; p = 0.13) bite point.

Variation in reaction forces generated at the TMJs

During premolar biting, all seven of the ALL-HUM models generated strongly compressive reaction forces at both TMJs (see Table 9), similar to the results for chimpanzees (Smith et al., 2015b). However, unlike in chimpanzees, M2 biting generated distractive (tensile) reaction forces at the working-side TMJ that would have “pulled” the mandibular condyle away from the articular eminence in five of the seven models. In order to remove distractive forces, these models required reductions in the muscle force applied to the balancing-side, which ranged from 5 to 15% (see Table 9). Interestingly, when loaded with chimpanzee muscle forces, all seven of the CHIMPED human models exhibit distractive forces in the working TMJ during M2 biting, with larger muscle force reductions required to eliminate the distraction (see below).

Biomechanical “performance” of human feeding

Structural stiffness of the human craniofacial skeleton

Direct comparisons of shape-related mechanical performance between our human FEMs and our previously analyzed chimpanzee FEMs (Smith et al., 2015a; Smith et al., 2015b) were permitted by the CHIMPED models. These comparisons reveal that the human craniofacial skeleton is less stiff and experiences von Mises strains that are elevated relative to those experienced by chimpanzees when subjected to identical loading conditions (Fig. 10). Several of the sampled locations were found to experience significantly higher magnitudes in humans during both P3 and M2 biting following the results of Holm-Bonferroni-corrected Mann-Whitney U tests (Table 10). These included the working nasal margin (Location 12), postorbital bars (Locations 4 and 5), working zygomatic root (Location 8), and the working dorsal orbital (Location 2). However, strains at the mid-zygomatic arches in humans were within the range observed for chimpanzees (which are extremely variable). Additionally, human zygomatic bodies were found to be structurally stiff, with significantly lower von Mises strain magnitudes than chimpanzees.

Figure 10 Line plots of von Mises microstrain generated during simulated biting in FEMs of humans and chimpanzees.

Strain data correspond to (A) left premolar (P3) and (B) left molar (M2) biting, recorded from 14 homologous locations in the CHIMPED variants of “extreme” and “average” modern human cranial FEMs. The gray region brackets the range of variation observed for chimpanzees by Smith et al. (2015b).

Table 10 Von Mises strain magnitudes: Human vs. Chimpanzee.

Results of pairwise comparisons (Mann-Whitney U-test) of von Mises strain magnitudes at the 14 locations examined between CHIMPED variants of human FEMs and data on chimpanzees from Smith et al. (2015b). Because of small sample sizes, the “exact” variant of p is reported (Mundry & Fischer, 1998). Comparisons that yielded significant results following Holm-Bonferroni correction are marked by asterisks. When significant, humans were found to exhibit the higher average value, with the exception of locations 13 and 14, where humans were found to exhibit significantly lower strain magnitudes.

Site	Bite	U	z	Exact p	
1. Dorsal interorbital	Premolar	9	−1.65	0.0967	
Molar	10	−1.50	0.1265	
2. Working dorsal orbital	Premolar	0	−2.93	0.0012*	
Molar	0	−2.93	0.0012*	
3. Balancing dorsal orbital	Premolar	4	−2.36	0.01401	
Molar	7	−1.93	0.0513	
4. Working postorbital bar	Premolar	0	−2.93	0.0012*	
Molar	1	−2.79	0.0023*	
5. Balancing postorbital bar	Premolar	0	−2.93	0.0012*	
Molar	0	−2.93	0.0012*	
6. Working zygomatic arch	Premolar	14	−0.93	0.3660	
Molar	14	−0.93	0.3660	
7. Balancing zygomatic arch	Premolar	14	−0.93	0.3660	
Molar	14	−0.93	0.3660	
8. Working zygomatic root	Premolar	0	−2.93	0.0012*	
Molar	0	−2.93	0.0012*	
9. Balancing zygo root	Premolar	18	−0.36	0.7308	
Molar	11	−1.36	0.1807	
10. Working infraorbital	Premolar	2	−2.64	0.0047*	
Molar	7.5	−1.86	0.0565	
11. Balancing infraorbital	Premolar	6	−2.07	0.03501	
Molar	12	−1.21	0.2343	
12. Working nasal margin	Premolar	0	−2.93	0.0012*	
Molar	1	−2.79	0.0023*	
13. Working zygomatic body	Premolar	0	−2.93	0.0012*	
Molar	1	−2.79	0.0023*	
14. Balancing zygomatic body	Premolar	0.5	−2.86	0.0017*	
Molar	1	−2.79	0.0023*	
Note:*

1 Result is significant at p ≤ 0.05.

Human bite force production and mechanical efficiency

Analysis of our CHIMPED human FEMs reveals that human crania are capable of generating bite forces with higher mechanical efficiency than chimpanzees (Fig. 11). Pairwise comparisons using the Mann-Whitey U test demonstrate that these differences are significant at both P3 (U = 1.5, z = −2.73, exact p = 0.003) and M2 (U = 1, z = −2.79, exact p = 0.002) bite points. However, unlike chimpanzees, all seven of the CHIMPED human models generated highly distractive (tensile) reaction forces at the working-side TMJ during molar biting. Therefore, molar biting in humans increases the risk of having the muscle resultant vector fall outside the triangle of support. To bring the joint back into compression, a reduction in balancing side muscle force of 15–30% was required (Table 11).

Figure 11 Biting efficiency: humans vs. chimpanzees.

Box-and-whisker plots show the minimum, first quartile, median, third quartile, and maximum biting efficiency, as quantified using the MA, in the CHIMPED variants of human cranial FEMs vs. chimpanzees at (A) premolar (P3) and (B) molar (M2) bite points. Chimpanzee data is from Smith et al. (2015b).

Table 11 Bite force production, biting efficiency, and joint reaction forces in the CHIMPED model variants of human crania.

Bite force (BF), mechanical advantage (MA), working-side TMJ reaction force (RF-WS), and balancing-side TMJ reaction force (RF-BS) for premolar and molar biting. All seven CHIMPED models generated highly distractive (tensile) reaction forces during molar loading that would have increased the chances of joint dislocation and/or injury. Therefore, balancing side muscle forces were iteratively reduced by 5% and re-run until distractive forces were eliminated. Bite forces and TMJ reaction forces are in Newtons (N).

Model	Muscle force	Premolar bite	Molar bite	
BF	MA	RF-WS	RF-BS	BF	MA	RF-WS	RF-BS	
GRGL	3,965	1,724	0.43	499.82	1,189.57	2,570	0.65	−208.16	1,113.51	
GRGL1	3,569					2,316	0.65	−31.26	841.64	
GRGL2	3,469					2,252	0.65	12.96	773.68	
BERG	3,637	1,720	0.47	405.08	935.03	2,599	0.71	−185.65	819.81	
BERG2	3,183					2,277	0.71	−6.72	560.17	
BERG3	3,092					2,213	0.71	29.07	508.24	
KSAN1	3,353	1,462	0.44	343.26	1,030.37	2,080	0.62	−187.95	975.38	
KSAN12	2,934					1,822	0.62	−0.30	687.33	
KSAN13	2,850					1,771	0.62	37.23	629.72	
KSAN2	2,804	1,272	0.45	311.70	821.79	1,895	0.68	−163.75	757.22	
KSAN22	2,454					1,658	0.68	−11.46	529.80	
KSAN23	2,384					1,610	0.68	18.99	484.32	
MALP	2,986	1,358	0.45	384.41	966.38	2,118	0.71	−203.31	963.66	
MALP2	2,613					1,851	0.71	−2.01	667.11	
MALP3	2,538					1,797	0.71	38.25	607.81	
TIGA	4,418	1,941	0.44	564.13	1,288.46	2,896	0.66	−107.59	1,143.16	
TIGA4	4,197					2,750	0.66	−13.27	997.33	
TIGA5	4,086					2,678	0.66	33.89	924.42	
WAFR	3,567	1,383	0.39	489.34	1,103.22	2,146	0.60	−61.09	1,006.50	
WAFR6	3,478					2,091	0.60	−24.01	946.69	
WAFR4	3,389					2,036	0.60	13.07	886.88	
Notes:

1 Model re-run using muscle forces reduced by 20% on the balancing side.

2 Model re-run using muscle forces reduced by 25% on the balancing side.

3 Model re-run using muscle forces reduced by 30% on the balancing side.

4 Model re-run using muscle forces reduced by 10% on the balancing side.

5 Model re-run using muscle forces reduced by 15% on the balancing side.

6 Model re-run using muscle forces reduced by 5% on the balancing side.

Discussion

In vitro validation

In order to validate the findings of our mechanical analysis, we compared in vitro bone strain in a cadaveric human head during simulated P3 biting to the results of a specimen-specific FEA. We found the results of our specimen-specific FEA corresponded quite well with in vitro data. In addition to the notable similarities in strain orientation at the 14 sampled locations, results of the regression analysis reveal that FEA can predict in vitro strain magnitudes with a high degree of accuracy (r2 values > 0.9). Similarly, Nagasao et al. (2005) were able to validate a dry bone human cranium with a high degree of accuracy (r2 = 0.989). However, these authors examined only 2 gage sites and they simulated biting by applying forces to teeth, thus omitting the impact of muscle loading. A greater number of sites were included in an analysis by Szwedowski, Fialkov & Whyne (2011), who found that their FEM results predicted in vitro data with an r2 of 0.73. Toro-Ibacache et al. (2015) also applied point loads to a cadaveric human head and validated strains at two locations in a specimen-specific FEM, finding broad similarities.

Although we found excellent correspondence between in vitro and in silico results, it is clear that FEA does incorporate error (see Table 6). This error was deceptively large at some “gage sites,” particularly in areas of low strain. For example, error for maximum principal strains at the balancing dorsal orbital (Location 3) was 80%, but this represents a difference between experimental and FEA results of only 2.67 microstrain (μɛ). Generally speaking, this is not a meaningful difference in the context of vertebrate feeding biomechanics, where some regions of the cranium can experience strain in the thousands of microstrain. However, some moderately strained areas exhibited high error percentages. In particular, the working infraorbital validated well for minimum principal strain, but error for maximum principal strain was nearly 50%. This discrepancy may be related to the morphology of the bone that forms the thin anterior wall of the maxillary sinus, which is susceptible to large modeling errors (Maloul, Fialkov & Whyne, 2011), or could be a result of simplifications to the thin bones of the nasal cavity (see Toro-Ibacache et al., 2015).

Mechanical variation

We found that the ALL-HUM models exhibited generally low levels of shape-related mechanical variation in strain magnitude and bite force production. Additionally, though some regions (e.g., the nasal margin) were found to exhibit large differences in strain magnitude, our human FEMs shared a common pattern of the spatial distribution of relatively high and low strain concentrations. These findings are similar to those of Smith et al. (2015b), who found broad similarities in strain patterning among on a sample of chimpanzee FEMs that differed notably in shape. Similarly, Toro-Ibacache, Zapata Muñoz & O’Higgins (2015) found broad similarities between two notably distinct humans cranial FEMs. Our finding that the ALL-HUM models exhibit low levels of mechanical variation supports the functional significance of the comparisons of shape-related mechanical performance made between our CHIMPED human FEMs and our previously analyzed chimpanzee FEMs (Smith et al., 2015a; Smith et al., 2015b), which focused purely on mechanical differences resulting from geometrical/architectural variation in the craniofacial skeleton.

Mechanical performance in humans and chimpanzee

Craniofacial strength: Is the human face weak?

Our results suggest that the modern human craniofacial skeleton is structurally less strong, in terms of resistance to masticatory stress, than that of chimpanzees when subjected to identical loading conditions (i.e., same properties and constraints, muscle forces scaled to model size). In the CHIMPED variants of our human FEMs, most of the locations analyzed experienced von Mises strain magnitudes that were elevated relative to chimpanzees, in particular the working nasal margin, the postorbital bars, the working zygomatic root, and the working dorsal orbital region. Exceptions to this pattern include the zygomatic arches, where strains were bracketed by the range of values seen in chimp FEMs, and the prominence of the zygomatic body (i.e., the “cheek bone”), which is apparently strong in modern humans.

During unilateral P3 biting, the nasal margin of modern humans experienced von Mises strains that were on average more than 350% greater than chimpanzees. Similarly, previous investigations identify the “root” of the nasal margin to be an area of high stress and strain during masticatory loading in humans (Endo, 1965; Endo, 1966; Arbel, Hershkovitz & Gross, 2000; Szwedowski, Fialkov & Whyne, 2011; Maloul et al., 2012). This region is often described as a pillar-like structure (Benninghoff, 1925; Bluntschli, 1926), or section of a frame-like structure (Görke, 1904; Endo, 1965; Endo, 1966), that resists mainly compression during anterior tooth biting. The results of our analysis are in general agreement with these findings, except that tension at the nasal margin was also found to be high in magnitude, indicating intense bending and shearing of the lower maxillary region during anterior tooth biting (see Figs. 7 and 9).

In addition to the nasal margin, the postorbital bars of the human FEMs were also found to experience highly elevated von Mises strain magnitudes compared to chimpanzees. However, adjacent regions, including the zygoma/zygomatic body (“cheek bone”) region and zygomatic arch, were found to be similar in strength to the lower end of the chimpanzee range. Mechanical analyses of Paranthropus boisei and Australopithecus africanus (Smith et al., 2015a) show a similar pattern of relatively low strains in the zygomatic body. Smith et al. (2015a) suggest that the structural strength of the zygomatic body in australopiths could be adaptively significant, offering as one possibility that it serves to reduce strains in the nearby zygomatico-maxillary suture. In pigs, it has been demonstrated that unfused sutures can fail at relatively modest stress levels (e.g., Popowics & Herring, 2007), so some bony facial regions may serve to shield nearby sutures from masticatory stresses rather than bone itself (Wang et al., 2012). Among smaller-faced modern human crania, the zygomatico-maxillary suture may be especially prone to experiencing relatively large masticatory stresses. In our FEMs, the largest strains in this region of the mid-face were generated medial to the zygomatico-maxillary suture. The location of these elevated strain magnitudes corresponds roughly to the location of facial fractures experienced commonly during physical altercations (Ellis, El-Attar & Moos, 1985). Facial fractures are also common at the postorbital bar, as opposed to the zygomatic body or zygomatico-maxillary suture, when the zygomatic body is exposed to traumatic blows (Ellis, 2012; Pollock, 2012). Therefore, it is possible that the strength of the human zygomatic body, and perhaps the relative weakness of the postorbital bar, is related to diverting stress from sutures that might otherwise fail under relatively lower stress magnitudes.

Like the zygomatic body (“cheek bone”) region, humans were found to exhibit lower average von Mises strains and markedly lower peak strains than chimpanzees at the mid-zygomatic arch, although human values were bracketed by the range of chimp values. This potentially reflects differences in arch length. Specifically, the size of the temporalis muscle, which is correlated with the area of the infratemporal fossa (Weijs & Hillen, 1984), is significantly reduced in humans compared to that of chimpanzees (Taylor & Vinyard, 2013). Demes & Creel (1988) show that the area of the infratemporal fossa is nearly half that of chimpanzees, meaning that the total length of the zygomatic arch is also reduced. Bone strain analyses demonstrate that the arch is subjected to sagittal bending, as well as torsion along its long axis (e.g., Hylander, Johnson & Picq, 1991; Hylander & Johnson, 1997; Ross, 2001; Ross et al., 2011). Predictions based on beam theory therefore suggest that a decrease in the length of the arch will lessen these bending and torsional moments, whereas a reduction in the height and/or breadth of the arch will weaken it under bending and shear, respectively.

Functional interpretations based on the morphology of the zygomatic arch are complicated by the fact that the temporalis fascia has been hypothesized to stabilize it from the inferiorly-directed pulling action of the masseter muscle (Eisenberg & Brodie, 1965). Curtis et al. (2011) tested this hypothesis using FEA and found that models that do not include the temporalis fascia will overestimate strains in the arch and surrounding regions, including the postorbital bar and infraorbital. However, they also found that their models lacking a fascia generated strains more similar in magnitude to those collected during in vivo experiments (Hylander, Johnson & Picq, 1991; Hylander & Johnson, 1997; Ross, 2001; Ross et al., 2011). Similarly, previous FEA studies on primate crania that have not included a modeled fascia (e.g., Ross et al., 2005; Ross et al., 2011; Strait et al., 2005) also find broad agreement with in vivo data. Therefore, we did not feel that it was necessary to include this structure in our FEMs. Importantly, Curtis et al. (2011) did not actually model the temporalis fascia, rather, they applied external forces along the margin of the attachment of the fascia. This procedure assumes that the load transferred to bone by the fascia is evenly distributed around its perimeter. However, the fascia is subjected to load by the inferiorly directed force produced by those temporalis fibers that arise off of the deep surface of the fascia. This force should elevate tension in the fascia along its superior margin (i.e., where it arises off of the superior temporal line) while reducing tension along its inferior margin (i.e., along the arch). This factor may mitigate the role of the fascia in resisting the contraction of the masseter muscle.

Although the brow ridges are not thought to play an important role in masticatory stress resistance (e.g., Picq & Hylander, 1989; Hylander, Johnson & Picq, 1991; Ravosa, 1991a; Ravosa, 1991b; Ravosa et al., 2000) it is interesting to note that our human FEMs experienced higher von Mises strain magnitudes than chimpanzees at all three of the supraorbital sites examined, particularly during premolar biting. Between the human and chimpanzee samples, differences were found to be greatest at the working and balancing dorsal orbitals, not the dorsal interorbital, supporting the idea that the brow ridge cannot be modeled as a bent beam (Picq & Hylander, 1989; see also Chalk et al., 2011). The fact that the smaller brows of humans experienced elevated strain magnitudes during biting could be interpreted as meaning that large brow ridges are an adaptation to resist masticatory loads. However, a wealth of experimental data on humans and non-human primate species has shown (e.g., Hylander, Johnson & Picq, 1991; Ravosa et al., 2000; Szwedowski, Fialkov & Whyne, 2011; Ross et al., 2011; Maloul et al., 2012) that strains along the supraorbital margin are relatively low during biting and chewing, which is supported by the results presented here. Therefore, it is more reasonable to interpret differences in supraorbital morphology between humans and chimpanzees as being related to some non-dietary function, and that the resulting increases in brow ridge strain among humans are experienced as a secondary byproduct. For example, Moss & Young (1960) suggest that a large separation is formed posterior to the orbits when brain size is small, forming a supraorbital ridge. When brain size is large, the frontal bone is more steeply inclined posterior to the orbits, forming a vertical forehead rather than a large torus. A byproduct of this missing bar of bone above the orbits among modern humans could be that strain magnitudes are mildly elevated in that region.

Overall, our findings show that the human craniofacial skeleton is weaker than that of chimpanzees when subjected to feeding loads. These findings support the hypothesis that dietary changes involving a shift to softer and/or more processed foods along the modern human lineage has led to masticatory gracilization and reduced structural strength of the bony facial skeleton (e.g., Lieberman et al., 2004). However, in their biomechanical analysis, Wroe et al. (2010) recently found that although the human cranium is less robust, it experiences low peak strains and an even distribution of facial strain magnitudes compared to extant apes and fossil australopith species. Differences between our results and those of Wroe et al. (2010) could reflect differences in the way muscle loads were applied to the models in each analysis and/or the manner in which models were constrained. For example, we applied both normal and tangential tractions over entire muscle areas using Boneload (Grosse et al., 2007), whereas Wroe et al. (2010) loaded their models with muscles modeled as straight pre-tensioned beam elements. However, we conducted a sensitivity analysis to explore this possibility further (see Supplemental Information) and found that these differences in methodology only resulted in small differences in strain magnitude at most locations across the craniofacial skeleton.

Another possible explanation for the differences between our study and the study by Wroe et al. (2010) relates to the magnitudes of the applied muscle forces. Wroe et al. (2010) subjected their FEMs to three sets of simulated biting on various teeth. In their first simulation of the three, FEMs were assigned a set of species-specific muscle forces (or muscle force estimates) from the literature. In a second simulation, models were scaled to the surface area of their chimpanzee model and re-loaded using chimpanzee muscle forces. Lastly, in the third simulation, models were scaled to the surface area of their chimpanzee model and loaded with muscle loads required to generate an equivalent bite force. In this third simulation, the high biting leverage offered by the retracted human face meant that the muscle forces required to generate a bite compared to the other hominoids examined were relatively low. Therefore, Wroe et al. (2010) concluded that the human facial skeleton may in fact be well-adapted to resist masticatory stresses generated during high magnitude biting. Importantly, however, mean element von Mises stresses were found to be relatively high in their human FEM during the second simulation, where FEMs were scaled to the same surface area and loaded with equivalent muscle forces. This is the most similar of their three scaling procedures to the scaling performed here (scaling muscle forces to model volume2/3), which we believe is the best means for removing the effects of size on comparisons of mechanical performance (e.g., Dumont, Grosse & Slater, 2009; Strait et al., 2010).

Bite force production and efficiency: are humans suited to produce large biting forces?

When analyzed using human bone and muscle properties (i.e., ALL-HUM models), our human FEMs produced bite forces of 333–507 N at the premolar (P3) and 496–756 N at the molar (M2). These results are similar to, but lower than, previous estimates of human bite force production using both 2D and 3D modeling techniques (e.g., Wroe et al., 2010; Eng et al., 2013). For example, using skeletal measurements and data on muscle cross-section, Eng et al. (2013) recently estimated that humans are capable of producing approximately 660–1106 N of M2 bite force, while Wroe et al. (2010) estimated a maximum unilateral M2 bite force of 1109–1317 N using FEA. However, our M2 bite force results are bracketed by bite force transducer data collected from various western populations, which range from approximately 368 N (Sinn, de Assis & Throckmorton, 1996) to around 911 N (Waltimo, Nystram & Kananen, 1994), although Inuit males have been shown to produce an average of 1277 N in M2 bite force (Waugh, 1937). Therefore, our results for bite force production lie within and do not exceed the known range of in vivo variation exhibited by recent human populations.

Because chimpanzees have absolutely and relatively larger jaw adductor muscles than humans (e.g., Taylor & Vinyard, 2013), it is no surprise that the chimp FEMs were capable of producing more forceful bites than our human FEMs when loaded with species-specific muscle forces (compare data in Table 9 to Smith et al. (2015b), Table 4). However, when loaded with muscle forces scaled to remove differences in size (as in the CHIMPED model variants), we found that humans are more efficient producers of bite forces, in terms of biting leverage, consistent with the findings of Wroe et al. (2010). Specifically, the MA for P3 biting in humans ranged 0.39–0.47, compared to 0.32–0.42 in chimpanzees (Smith et al., 2015b), with only two chimps overlapping the human range. Humans were found to exhibit even more elevated leverage during M2 biting (0.60–0.71), with only one individual overlapping the chimpanzee range (0.49–0.61). When comparing these data using the Mann-Whitey U test, humans were found to be significantly more efficient at producing bite forces at both mesial and distal bite points. The CHIMPED humans were even found to exhibit a biting efficiency similar to that observed in australopiths (Smith et al., 2015a). In fact, P3 MA in P. boisei (0.40) and A. africanus (0.41) were near the lower end observed in humans. The FEM of A. africanus also generated M2 bites with similar efficiency (0.62) to humans, whereas P. boisei produced more mechanically efficient (0.75) molar bites (Smith et al., 2015a).

Our data on bite force efficiency in humans support previous findings that have demonstrated the mechanical advantage of modern human bony facial architecture compared to both non-modern humans and non-human primate species (e.g., Spencer & Demes, 1993; O’Connor, Franciscus & Holton, 2005; Lieberman, 2008; Lieberman, 2011; Wroe et al., 2010; Eng et al., 2013). Using estimates of muscle leverage from 2D measurements (Lieberman, 2008; Lieberman, 2011), humans have been shown to achieve high biting leverage through a marked degree of facial retraction (orthognathism), which reorients the muscles of mastication relative to the tooth rows. As noted above, we found that our human FEMs produced bite forces with leverage ratios similar to those observed in A. africanus and P. boisei (Smith et al., 2015a). However, australopiths achieve high biting leverage through an anterior positioning of the chewing muscles relative to the tooth rows (Rak, 1983; Strait et al., 2009; Strait et al., 2010; Smith et al., 2015a). In humans, the midfacial region is “tucked” beneath the anterior cranial fossa (Lieberman, McBratney & Krovitz, 2002; Lieberman et al., 2004; Lieberman, 2008; Lieberman, 2011), which similarly places bite points in a position that offers higher mechanical advantage to the jaw adductors.

Although the human cranium can theoretically produce mechanically efficient bite forces, the production of unilateral molar (M2) bite force is limited by the risk of TMJ distraction, as predicted by the constrained lever model (Greaves, 1978; Spencer, 1998; Spencer, 1999). Specifically, we found that all seven of the CHIMPED human FEMs experienced a highly distractive (tensile) reaction force at the working-side joint during molar biting. These forces have the effect of “pulling” the mandibular condyle from the jaw joint, increasing the risk of joint dislocation (Spencer, 1998; Spencer, 1999). As noted in the introduction, the soft tissues of the mammalian jaw joint are well suited to resist compressive joint reaction forces, but are poorly configured to resist distractive joint forces that “pull” the mandibular condyle from the cranial base (Greaves, 1978; Spencer, 1998; Spencer, 1999). In contrast, only one of the six chimpanzee FEMs analyzed by Smith et al. (2015a) generated a tensile force at the working TMJ, and this reaction was only very weakly tensile (12.7 N). Similarly, Smith et al. (2015b) found that their FEMs of P. boisei and A. africanus lacked working-side distraction and were able to produce “stable” bites on both the premolars and molars, offering these species the ability to produce maximally forceful molar bites with limited risk of causing pain and/or damage to the TMJ capsule.

Interestingly, when loaded with human muscle forces (i.e., ALL-HUM), two of the human FEMs (TIGA and WAFR) were capable of maintaining weakly compressive reaction forces at both TMJs during molar biting. Additionally, balancing side force reductions required to eliminate distraction in the remaining models were proportionately less (5–15%) than when applying chimpanzee forces (15–30%). Comparisons of the muscle loads applied to the models and their force ratios in the ALL-HUM and CHIMPED models (see Tables 9 and 11) reveal that chimpanzees devote a higher proportion of muscle strength to anteriorly-positioned muscle compartments (superficial masseter and anterior temporalis) compared to more posteriorly-positioned ones (deep masseter and medial pterygoid). Therefore, it is tempting to suggest that changes in human jaw muscle force ratios may have coincided with the retraction of the lower face during human evolution in order to reduce the risk of TMJ distraction. Likewise, if the repositioning of cranial elements for reasons other than food processing (Lieberman, 2008; Zink & Lieberman, 2016) led to an increase in biting efficiency but the generation of working side joint distraction during molar biting, the overall reduction of chewing muscle size in Homo could also be viewed as a result of positive selection rather than relaxed selection so as to lessen these distractive forces.

Our findings that humans are limited in their ability to produce forceful unilateral molar bites are supported by data on bite force and muscle activity in humans. Spencer (1995) and Spencer (1998) tested some predictions of the constrained lever model and found that humans produced bite forces that increased as the bite point moved from the incisors to the first molar. However, moving from M1 to M3, bite forces were found to decrease as a result of the decreasing balancing force muscle recruitment required to avoid joint distraction. Spencer (1995) also notes that most of the participants (8 of 10) in his analysis reported pain near the working-side TMJ when biting forcefully using the back molars. In addition to this study, Hylander (1977) suggests that specialized anterior tooth biting and increased masticatory muscle leverage may be related to the high incidence of third molar reduction and agenesis among modern Inuit due to the increased risk of distraction when biting on these teeth, although the results of our single pre-historic Arctic FEM (TIGA) provide no support for this hypothesis. Similarly, Spencer (2003) demonstrates that seed predating New World primates with adaptations for increased anterior bite force have relatively small third molar roots.

As discussed above, Wroe et al. (2010) analyzed human feeding biomechanics within a comparative context. One of the principal findings of their analysis, supported by the data presented here, is that humans are capable of generating bite forces with higher mechanical efficiency than chimpanzees. Wroe et al. (2010) use this as evidence to argue that human craniofacial evolution may have been influenced by selection for powerful biting behaviors. However, the results of this study showing the comparative weakness of the human cranium combined with the increased risk of jaw joint distraction during molar biting leads us to interpret the increased biting leverage exhibited by humans, which is particularly high among recent populations (Spencer & Demes, 1993; O’Connor, Franciscus & Holton, 2005), to be a byproduct of human facial orthognathism, which may be at least partly related to facial size reduction. Human facial flatness may also have been acquired through selection for some non-dietary function. For example, Lieberman (2008) and Lieberman (2011) suggests that the marked degree of facial retraction exhibited by modern human crania could be related to changes in brain size and cranial base flexion. However, Ross (2013) shows that basicranial flexion cannot produce significant facial retraction on its own. Alternatively, Holton et al. (2010) propose that dietary shifts leading to reduced facial strain magnitudes among early human species may have led to reduced facial growth and earlier fusion of the maxillary sutures, and thus smaller and more retracted facial skeletons.

Although the majority of the morphological and mechanical evidence is not consistent with the hypothesis that the human masticatory apparatus has experienced recent selection for high magnitude biting, the results of our analysis cannot reject the hypothesis that, in addition to changes in diet and tool use, increases in muscle force efficiency during human evolution could have led to relaxed selection for large chewing muscle size and reductions in facial size (Wroe et al., 2010) or that humans benefited from increased biting leverage when using submaximal forces by exerting less energy per bite. Our results for premolar biting leverage also do not conflict directly with the hypothesis that anterior tooth biting could have been selectively important in humans. However, the reduced size of the premolar teeth in humans increases the risk of tooth crown fracture (Constantino et al., 2010). Therefore, studies on premolar size and strength are not consistent with the hypothesis that humans are particularly well adapted for forcefully loading their anterior teeth, but such studies have yet to be conducted on incisors or canines, which are the more likely to be used during paramasticatory activities. For example, Hylander (1977) identifies features of the modern Inuit craniofacial skeleton that he argues to be adaptations for powerful biting behaviors using the incisors, although our single pre-historic Arctic FEM (TIGA) was not found to be exceptional in this regard. Additionally, Spencer & Ungar (2000) show that incisor bite force leverage varies in relation to the intensity of incisor tooth use among some Native American populations. Similarly, it is possible that differences in anterior tooth use among “archaic” members of the genus Homo are reflected in mechanical differences between the species. In particular, the Neanderthals (H. neanderthalensis) exhibit a number of derived characteristics hypothesized to be adaptations for forceful incisor biting (e.g., Brace, 1962; Smith, 1983; Trinkaus, 1983; Trinkaus, 1987; Rak, 1986; Demes, 1987). Notably, Spencer & Demes (1993) show that Neanderthals exhibit high incisor bite force leverage relative to H. heidelbergensis (but not modern H. sapiens). In order to maintain functional use of the posterior dentition (i.e., avoid TMJ distraction), Spencer & Demes (1993) further show that the molar tooth row in Neanderthals was anteriorly shifted, resulting in the characteristic retromolar gap.

Data on enamel thickness seemingly contrasts with the hypothesis that humans have experienced relaxed selection for powerful biting behaviors. Specifically, a number of studies find that recent human populations exhibit thick molar enamel (e.g., Martin, 1983; Martin, 1985; Olejniczak et al., 2008; Smith et al., 2006; Vogel et al., 2008), which has been interpreted as a primitive retention. However, notwithstanding disagreements over the significance of enamel thickness (Grine, 2005), Smith et al. (2012) recently show that “thick” molar enamel in humans is primarily the result of small coronal dentine areas. They found that enamel area in humans is reduced, but there was a disproportionately large reduction in dentine to enamel as human teeth were evolving smaller size, resulting in a relatively “thick” enamel cap. Thus, Smith et al. (2012) argue that the dichotomy between thick and thin enamel is an oversimplification.

Conclusions

We examined the biomechanical consequences of human masticatory gracilization and intraspecific variation within the constrained lever model of feeding biomechanics (Spencer, 1999) and tested the hypothesis that the human face is well configured to generate and withstand high biting forces relative to chimpanzees. We found that our biomechanical models of human crania were, on average, less structurally stiff than the crania of chimpanzees when assigned equivalent bone properties, constraints, and physiologically-scaled muscle forces. These results are consistent with the facial reduction exhibited by modern humans. We also found that modern humans are efficient producers of bite force, consistent with previous analyses (Spencer & Demes, 1993; O’Connor, Franciscus & Holton, 2005; Lieberman, 2008; Lieberman, 2011; Wroe et al., 2010; Eng et al., 2013), but that distractive (tensile) reaction forces are generated at the working (biting) side jaw joint during M2 biting. In life, such a configuration would have increased the risk of joint dislocation and constrained the maximum recruitment levels of the masticatory muscles, meaning that the human cranium is poorly suited to produce forceful unilateral molar bites. Our results do not conflict directly with the hypothesis that premolar biting could have been selectively important in humans, although the reduced size of these teeth in humans has been shown to increase the risk of tooth crown fracture. We interpret our results to suggest that human craniofacial evolution was probably not driven by selection for high magnitude biting, and that increased masticatory muscle efficiency in humans is likely to be a byproduct of selection for some non-dietary function (Lieberman, 2008) or perhaps related to reduced masticatory strain and sutural growth restrictions (Holton et al., 2010).

Our results provide support for the hypothesis that a shift to the consumption of less mechanically challenging foods and/or the innovation of extra-oral food processing techniques (e.g., stone tool use, cooking) along the lineage leading to modern Homo sapiens relaxed the selective pressures maintaining features that favor forceful biting and chewing behaviors, including large teeth and robust facial skeletons, leading to the characteristically small and gracile faces of modern humans (e.g., Brace, Smith & Hunt, 1991; Wrangham et al., 1999; Lieberman et al., 2004; Ungar, Grine & Teaford, 2006; Wood, 2009). To contribute to our further understanding, future studies should aim to identify the ecological changes that may have led to the emergence of such shifts in dietary behavior. Were these changes initiated by changes in climate, competition, resource availability, or some combination of these factors? To what extent is craniofacial gracilization part of a general pattern of skeletal gracilization in humans (Ruff et al., 1993; Ruff et al., 2015; Chirchir et al., 2015; Ryan & Shaw, 2015)? These questions will be addressed by gaining further insight into the dietary ecology and feeding adaptations of species near the origins of the modern human lineage through work on biomechanics, paleoecology, archaeology, bone chemistry, and dental wear, each of which inform key components necessary to obtaining a more complete understanding of human craniofacial evolution.

Supplemental Information

Supplemental Information 1 Supplementary Methods.

Click here for additional data file.

Supplemental Information 2 Cortical bone mechanical properties collected from two cadaveric human specimens.

E3 and v23 refer to the elastic (Young’s) modulus and Poisson’s ratio in the axis of maximum stiffness, respectively. For modulus, factor and temperature data were used to distribute regionally variation mechanical properties throughout each of the ALL-HUM models (see Main Text).

Click here for additional data file.

Supplemental Information 3 Strain and strain energy density results from simulated premolar bites.

Maximum principal strain (MaxPrin), minimum principal strain (MinPrin), strain mode (Mode), maximum shear strain (Shear), von Mises strain, and strain energy density (SED) generated during simulated premolar (P3) biting in the ALL-HUM variants of “extreme” and “average” modern human cranial FEMs. Site numbers follow Fig. 4.

Click here for additional data file.

Supplemental Information 4 Strain and strain energy density results from simulated molar bites.

Maximum principal strain (MaxPrin), minimum principal strain (MinPrin), strain mode (Mode), maximum shear strain (Shear), von Mises strain, and strain energy density (SED) generated during simulated molar (M2) biting in the ALL-HUM variants of “extreme” and “average” modern human cranial FEMs. Site numbers follow Fig. 4.

Click here for additional data file.

Supplemental Information 5 Beam forces used in sensitivity analysis.

Total muscle forces, beam count, and force per beam for each muscle group assigned to the GRGL model in the sensitivity analysis. Forces are in Newtons (N).

Click here for additional data file.

Supplemental Information 6 In vitro loading of human cranium.

Illustration of the loading apparatus constructed for the current analysis within the INSTRON loading machine during loading of the left P3.

Click here for additional data file.

Supplemental Information 7 Transparent view of the model under in vitro validation.

The surface model is shown in the position it was constrained during muscle loading, as in Fig. S1.

Click here for additional data file.

Supplemental Information 8 Principal strain orientations recorded during validation analysis: Sites 1, 2, and 3.

Purple lines represent the orientation of minimum principal strain (compression), which is 90° to orientation of maximum principal strain. Black circles represent location of strain gages at the dorsal interorbital (site 1), working-side dorsal orbital (site 2), and balancing-side dorsal orbital (site 3) during in vitro bone strain analysis. Three lines through each gage correspond to the orientation of principal strains during the in vitro loading analysis which were recorded in degrees relative to the A element of the gage.

Click here for additional data file.

Supplemental Information 9 Principal strain orientations recorded during validation analysis: Sites 8, 10, and 12 strain.

Blue lines represent the orientation of maximum principal strain (tension). Purple lines represent the orientation of minimum principal strain (compression), which is 90° to orientation of maximum principal strain. Black circles represent location of strain gages at the working-side zygomatic root (site 8), working-side infraorbital (site 10), and working-side nasal margin (site 12) during in vitro bone strain analysis. Three lines through each gage correspond to the orientation of principal strains during the in vitro loading analysis which were recorded in degrees relative to the A element of the gage.

Click here for additional data file.

Supplemental Information 10 Principal strain orientations recorded during validation analysis: Sites 9 and 11.

Blue lines represent the orientation of maximum principal strain (tension). Purple lines represent the orientation of minimum principal strain (compression), which is 90° to orientation of maximum principal strain. Black circles represent location of strain gages at the balancing-side zygomatic root (site 9) and balancing-side infraorbital (site 11) during in vitro bone strain analysis. Three lines through each gage correspond to the orientation of principal strains during the in vitro loading analysis which were recorded in degrees relative to the A element of the gage.

Click here for additional data file.

Supplemental Information 11 Principal strain orientations recorded during validation analysis: Sites 4, 6, and 13.

Blue lines represent the orientation of maximum principal strain (tension). Purple lines represent the orientation of minimum principal strain (compression), which is 90° to orientation of maximum principal strain. Black circles represent location of strain gages at the working-side postorbital bar (site 4), working-side zygomatic arch (site 6), and the working-side zygomatic body (site 13) during in vitro bone strain analysis. Three lines through each gage correspond to the orientation of principal strains during the in vitro loading analysis which were recorded in degrees relative to the A element of the gage.

Click here for additional data file.

Supplemental Information 12 Principal strain orientations recorded during validation analysis: Sites 5, 7, and 14.

Blue lines represent the orientation of maximum principal strain (tension). Purple lines represent the orientation of minimum principal strain (compression), which is 90° to orientation of maximum principal strain. Black circles represent location of strain gages at the balancing-side postorbital bar (site 5), balancing-side zygomatic arch (site 7), and balancing-side zygomatic body (site 14) during in vitro bone strain analysis. Three lines through each gage correspond to the orientation of principal strains during the in vitro loading analysis which were recorded in degrees relative to the A element of the gage.

Click here for additional data file.

Supplemental Information 13 The GRGL finite element model showing constraints and muscle loads applied following Wroe et al. (2010).

We compared two variants of this “beamed” model to our original “boneloaded” model, one that only included muscle beams for the anterior temporalis, superficial masseter, deep masseter, and medial pterygoid muscles (A) and a second that also included that posterior temporalis (B).

Click here for additional data file.

Supplemental Information 14 Results of sensitivity analysis: color maps of von Mises strain magnitudes.

Panels show strain distributions during premolar (P3) biting in the (A) original “boneloaded” ALL-HUM model, (B) “beamed” model lacking a posterior temporalis, and (C) “beamed” model including a posterior temporalis. Scales are set to range from 0–300 μɛ White regions exceed scale.

Click here for additional data file.

Supplemental Information 15 Results of sensitivity analysis: line plot of von Mises strain.

Plot shows the microstrain generated during simulated premolar (P3) biting, recorded from 14 identical brick elements across the craniofacial skeletons of our original “boneloaded” model, a “beamed” variant with muscle forces and constraints modeled following Wroe et al. (2010), and a third model analyzed following Wroe et al. (2010) but with the addition of the posterior temporalis (PT) muscle.

Click here for additional data file.

We thank Gisselle Garcia-Pack and Kristen Mable of the AMNH for access to human skeletal collections. We also thank Tim Ryan and Tim Stecko of the Center for Quantitative Imaging at Penn State for assistance in acquiring CT image data of modern human crania. Lastly, we thank the Academic Editor and the three reviewers who provided helpful comments and critique on an earlier draft of this paper.

Additional Information and Declarations

Competing Interests

Author Contributions

Data Deposition

The authors declare that they have no competing interests.

Justin A. Ledogar conceived and designed the experiments, performed the experiments, analyzed the data, wrote the paper, prepared figures and/or tables, reviewed drafts of the paper.

Paul C. Dechow conceived and designed the experiments, wrote the paper, reviewed drafts of the paper.

Qian Wang conceived and designed the experiments, wrote the paper, reviewed drafts of the paper.

Poorva H. Gharpure collected data on bone mechanical properties, wrote the paper, reviewed drafts of the paper.

Adam D. Gordon performed statistical analyses, wrote the paper, reviewed drafts of the paper.

Karen L. Baab performed statistical analyses, wrote the paper, reviewed drafts of the paper.

Amanda L. Smith wrote the paper, reviewed drafts of the paper.

Gerhard W. Weber conceived and designed the experiments, wrote the paper, reviewed drafts of the paper.

Ian R. Grosse conceived and designed the experiments, wrote the paper, reviewed drafts of the paper.

Callum F. Ross conceived and designed the experiments, wrote the paper, reviewed drafts of the paper.

Brian G. Richmond conceived and designed the experiments, wrote the paper, reviewed drafts of the paper.

Barth W. Wright conceived and designed the experiments, wrote the paper, reviewed drafts of the paper.

Craig Byron wrote the paper, reviewed drafts of the paper.

Stephen Wroe wrote the paper, reviewed drafts of the paper.

David S. Strait conceived and designed the experiments, wrote the paper, reviewed drafts of the paper.

The following information was supplied regarding data availability:

Raw data on strain from finite element models and bone mechanical properties are provided in the Supplemental Materials.

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
