# Peer review of "Human feeding biomechanics: performance, variation, and functional constraints"

_PeerJ, doi:10.7717/peerj.2242_

## Round 0.1 · original submission · Major Revisions

With 3 reviews in hand, all of which find your study to be very interesting indeed, I am recommending acceptance of your study pending rather substantial revisions.

I apologize for the delay, but the third reviewer asked for more time and just delivered a helpful review. Reviewer 1 identifies himself and makes a number of insightful comments and and reasonable requests with respect to "experimental design" and validation (be sure to read the annotations on the pdf too). Feel free to contact this reviewer directly if you have additional questions.

Although the 2nd and 3rd reviewers (both anonymous) checked the "minor revisions" box, attention to their comments in the "experimental design" section will require a thoughtful and detailed response (e.g., estimated muscle forces).

All 3 reviewers see merit in this study, as do I, and I hope you will find the reviews helpful as your prepare your revision.

·

Basic reporting

In this study, Ledogar and colleagues conducted a set of finite element simulations of premolar and molar biting in Homo sapiens crania and compared the resulting strain and mechanical advantage to those of Pan troglodytes. In general the paper is well-written and most of the methodology is explained with sufficient detail.

Experimental design

There is some discrepancy in the number of anatomical landmarks reported in the text versus the data tables. Additional implications of the collection of paired/symmetrical landmark data on the ordination of specimens in the PCA morphospace should be addressed. The sampling nodes on FE models from which strain values are reported should be more clearly defined. The claim of FE model validity using comparisons of in silico and in vitro data needs to be better explained, perhaps using quantitative/statistical assessments. Last but not least, the number and type of finite elements used, number of nodes, and degrees of freedom in the analyzed models could not be located. Please identify the location of that data more clearly, or add them in.

Validity of the findings

The authors found that human crania are generally weaker than chimp crania, but the former exhibits higher mechanical efficiency compared to the latter. Furthermore, molar biting using bilaterally maximal muscle contractile forces generate potentially unstable tensile forces at the TMJ in human models but not as much in chimp models. The discussion covers the findings in the context of previous hypotheses about cranial shape changes in anatomically modern humans relative to other homonids and great apes.

Additional comments

This is a well-constructed study with consideration of intraspecific variation and validation in masticatory simulations employing finite element analysis. The quantitative analyses and comparisons of simulation output data and reduced reliance on qualitative visual examination of strain maps relative to similar studies in the literature demonstrates the increasing explanatory power of such an approach in comparative anatomy.

Reviewer 2 ·

Basic reporting

Overall, this is a well-written manuscript. There are, however, and few areas that need clarification.

1) Sentence starting line 129: “If gracilization in Homo is a consequence of the removal of selection pressure to maintain and resist high magnitude or repetitive bite forces, then human feeding systems should not produce bite force efficiently and the cranium should be structurally week…” I agree with the overall sentiment of this sentence, but find the exact wording problematic. There is no reason why reduced selection could not result in a more efficient bite force system if it is a byproduct of selection for a less prognathic face due to something else (speech production, etc.)

2) Sentence starting on line 180: “Thus, although one might expect that a bite on a distal tooth would produce an elevated bite force due to a short load arm….” A shorter load arm would mean that LESS force is needed to fracture a given food item.

3) Line 416 mentions the “human models” implying that the data are from the “All-human” skulls. The legend for figure 9 indicates, however, that the data are from the “Chimped” models.

4) Tables: To improve clarity, many of the tables should include measurement units (when applicable) and require a bit more description of the data presented. For example, in Table 1 it is not at all clear what the #s represent. Another example is Table 3 – what are the units of measurement? How does “Distance from centroid”, differ from the “distance expressed in units of mean distance” (i.e. the data in parentheses)? Yet another example is Table 5 – what is the volume that is provided? Cross-sectional area? Total muscle volume?

5) Figures: Figure 2 – Although the legend mentions eclipses, there are none shown in the data plots.


Minor Typos:

Line 151: “hominids” should be “hominins”

Line 303: “Additionally, strain mode, the absolute value….” This sentence is redundant since strain mode was mentioned in the previous sentence (Line 300)

Line 353: The regression equations presented for the maximum and minimum principle strain are the same. While this might be a coincidence, it is worth a quick check to make sure that this is not a typo.

Line 355: Error was introduced into the “FEM” not the “FEA”???

Experimental design

While PeerJ does not evaluate based on impact, it should be noted that this study addresses an important paradox of human facial structure - while the size of the masticatory apparatus is reduced relative to other hominins (indicating relaxed selection for high chew effort) it is also more efficiently designed for producing bite forces. Thus the results of this study will be of great interest to any researcher studying the evolution of hominin facial form and masticatory performance.

Having said that, I do have a few comments/questions about the experimental design and methods.

1) It is not clear how applying chimp material properties (and forces) to a human skull is informative to the question being asked. As I am sure the authors are well-aware, the overall material properties of a complex structure such as the head is more than its component parts, and in FEM validating the model is of great import. Validating the ‘Chimped’ skull model is impossible and thus presents unknown error into the analysis. Why not simply focus on the ‘all human’ data and compare it to the chimp data presented in Smith et al 2015b? (Based on a very quick survey of Smith et al. 2015, it appears that similar methods were used in that study.)

2) Starting at line 260, it is not quite clear how the muscle forces were calculated/applied to the ‘extreme’ skulls. Why were forces calculated for the ‘average’ skull, but not the other skulls? How exactly were the forces scaled for the ‘extreme’ skull (and why were the values not simply calculated as they were for the ‘average’ skull?) Table 5 does not clarify my questions as it is not clear what the “volume” and “volume2/3” data represent. Are these total muscle volumes? Cross-sectional areas?

3) Line 242 states that fresh-frozen skulls were used for material property determination. Freezing has a pronounced effect on material properties – how might this affect the results of study? Or is this accounted for in the methods used (resistance to ultrasonic wave propagation)?

4) The FEM model was validated only for P3 loadings. Is there any reason to suspect that the error would be different if the M2 loadings were compared?

Validity of the findings

No comments. Validity of findings is sound (but see comments on experimental design).

Reviewer 3 ·

Basic reporting

This is a nice paper that includes a lot of data and effort by a number of authors. It's strengths include a consideration of shape variation in human crania and how extremes of human shape relate to mechanical performance. Another strength is the ex vivo validation of a human FEM that overall shows reasonable to good congruence between ex vivo and in silico strain magnitudes and orientation. It's also interesting and helpful for the field to see a thorough comparison of how and why the output of FE models of the same species (in this case H. sapiens) can vary.

Experimental design

I have no issues with the experimental design, with the exception of one question that I would like the authors to address. The human variant muscle forces were calculated by scaling average muscle forces up or down based on differences in model volume. This in itself is not problematic, but, muscle forces for the chimped models were estimated based on the scaling relationships devised by Smith et al. This left me wondering, how would the chimp forces calculated via the Smith et al. scaling relationship differ from chimp forces calculated via the volumetric scanning approach? Or vice versa, how would estimated human muscle forces differ if calculated via the chimp scaling relationship instead of via volumetric scaling? This is especially important as in the Discussion and the SI the authors identify differences in muscle forces as being one of the most important factors responsible for the differences between the results of Wroe et al. 2010 and the results presented here. Can the authors please address this point.

Validity of the findings

The conclusions on human craniofacial strength and efficiency are supported by the data presented here, and an extensive discussion is provided. At a number of points in the paper the authors refer to selection for non-dietary function, but do not elaborate on this further. Maybe they don't want to enter the realm of speculation, but can the authors provide at least some indication of what they mean by this? And what role does craniofacial sexual selection have to play in this? This is not my area, but there seems to be no mention of sexually selected craniofacial traits this the paper - even the discussion is solely focused on mechanical adaptations.

Additional comments

I spotted a few typos:
Line 131 – missing ‘as’
Line 179 – ‘reduces’
SI
Top page 7 – polygon misspelt.

One further question, why was the posterior temporalis omitted from the main analysis presented here? It was included in Wroe et al. 2010 (and is considered in the SI when the Wroe et al loading conditions are compared to the models used here).

---

## Round 0.2 · accepted · Accept

The reviewers clearly appreciate your efforts to respond to their queries and constructive criticism. Two the reviewers offer some minor points worthy of consideration when you submit the final final version of your ms to production. Please speak to the production unit about incorporating these requests. I am delighted to recommend acceptance of your study. Congratulations.

·

Basic reporting

Figure 2: the "red ellipses" appear rather faint on my screen; thicken or use a brighter color to show them more clearly.

Experimental design

L322 in tracked changes document: please explain how the surrogate mandibles for BERG and KSAN2 were scaled (linear? volume?).

Validity of the findings

No Comments

Additional comments

The authors have addressed all of my queries in their response and revised manuscript. I included a few minor suggestions to clarify two aspects of figure/methodology presentation. I think the overall approach and results are more clearly explained in this revised submission, and would recommend publication following those very minor revisions. Congratulations on a valuable contribution to FEA in comparative biology and human craniofacial biomechanics.

Reviewer 2 ·

Basic reporting

The revisions made by the authors greatly improve clarity of the manuscript. I have only two very small comments (personal preferences), neither of which are substantial. Publication should not be impeded should the authors choose not to make the suggested changes.

1) Abstract - "An alternative hypothesis states that the modern human face is adapted to generate and withstand high biting forces." I suggest adding something that very briefly explains why this is a valid hypothesis. The first part of the abstract well-describes the conventional view of the human masticatory system being weak. This description is then followed by a blanket statement (see quoted sentence above) that there is an alternative hypothesis. An uninformed reader will not be able to discern WHY this is an important alternative hypothesis to test. Because many people rely on abstracts to determine whether or not to read a paper, it is my opinion that the authors are doing themselves a disservice by not clearly articulating (within the abstract) the masticatory "paradox" presented by human craniofacial form.

2) Perhaps it is my computer/printer, but I have difficulty viewing the ellipses on Figure 2 (they are nearly invisible). I suggest making the ellipses darker/bolder.

Experimental design

No comments.

Validity of the findings

No comments.

Additional comments

The authors have sufficiently addressed my questions/comments on their previous manuscript submission. The revised manuscript is much improved.

Reviewer 3 ·

Basic reporting

The authors have comprehensively addressed my comments and made appropriate modifications to the manuscript. I have no further comments.

Experimental design

No comment - see above

Validity of the findings

No comment - see above

Additional comments

No comment - see above